# Gaussian Process Spatial Clustering

## Abstract

Spatial clustering is a common unsupervised learning problem with many applications in areas such as public health, urban planning or transportation, where the goal is to identify clusters of similar locations based on regionalization as well as patterns in characteristics over those locations. Unlike standard clustering, a well-studied area with a rich literature including methods such as K-means clustering, spectral clustering, and hierarchical clustering, spatial clustering is a relatively sparse area of study due to inherent differences between the spatial domain of the data and its corresponding covariates. In the case of our motivating example, the American Community Survey dataset, spatial differences in census tract regions cannot be directly compared to differences in participant survey responses to indicators such as employment status or income. As such, in this paper, we develop a spatial clustering algorithm called Gaussian Process Spatial Clustering (GPSC), which clusters functions between data leveraging the flexibility of Gaussian processes and extends it to the case of clustering geospatial data. We provide theoretical guarantees and demonstrate its capabilities to recover true clusters in several simulation studies and a real-world dataset to identify clusters of tracts in North Carolina based on socioeconomic and environmental indicators associated with health and cancer risk.

## 1 Introduction

There is growing research suggesting that socioenvironmental factors can play a key role in affecting health outcomes, potentially contributing to health disparities in marginalized groups, and may even predictably impact outcomes at the molecular level with diseases such as cancer (Lord et al., 2022; Larsen et al., 2020). However, identifying areas of such risk can be a difficult task. In the community-wide socioeconomic and environmental indicators dataset, the spatial locations of North Carolina census tracts were paired with socioeconomic data from the American Community Survey (ACS, 2014) from 2014 chosen to reflect socioeconomic advantage and disadvantage (Palumbo et al., 2016), as well as environmental pollution data from the U.S. Environmental Protection Agency (EPA) National Air Toxics Assessment (NATA, NAT (2014); Larsen et al. (2020)). This then poses the problem: How can geographically spread NC census tracts be clustered together based on risk factors including socioeconomic indicators and environmental pollution? North Carolina is known to be an ethnically diverse state (Emerson et al., 2020), with a wide range of spatially dependent differences in socioeconomic status such as access to healthcare, poverty rates, and education, while meaningful clusterings must take into consideration all these differences (Emerson et al., 2020). A standard clustering algorithm applied to the data collected from the patients in each tract or to the environmental variables alone fails to necessarily capture the significant spatial dependence inherent in the data collected in the studies. This problem is known as spatial clustering or geospatial clustering (Aldstadt, 2010).

In spatial clustering, the goal is to identify clusters of similar locations based on regionalization, as well as patterns in characteristics over those locations. Clustering of geospatial data is a common unsupervised learning problem with many applications to areas, e.g., public health, urban planning, or transportation, where geography plays an essential role.

Furthermore, spatial data, also known as geospatial data, is commonly characterized by having a distinct geographic component (Kisilevich et al., 2009). Unlike traditional data that only include observations as a single set of features $x$, spatial data may be considered as a vector $[s, x]$, where $s \in \mathbb{R}^2$ represents the

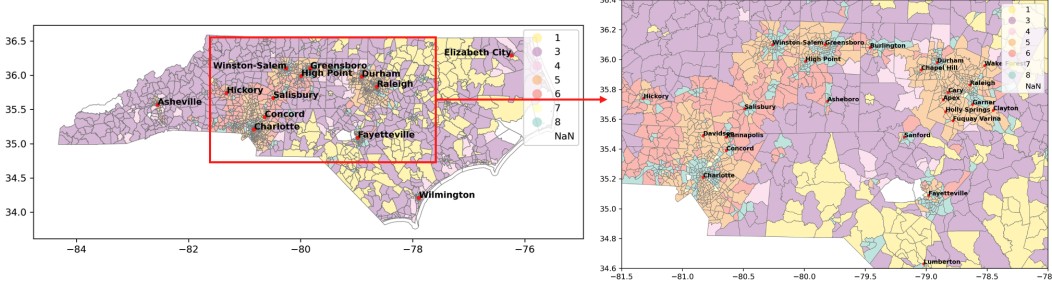

Figure 1: Distribution of socioeconomic and environmental advantage-disadvantage latent class in NC.

spatial location of the observation and $x \in \mathbb{R}^p$ is the set of features or covariates. The analysis of such spatial datasets poses challenges, such as accurately capturing the relative effects between the spatial and covariate domains (Kisilevich et al., 2009). Importantly, geographically close areas may still have very different patterns of characteristics, while separated areas may share similarities and constitute a single functional cluster. Together, this can pose challenges to traditional clustering methods that equally treat the separate domains inherent to geospatial data such as K-means, as the geographic locations of distinct clusters may be well mixed, or the measurements themselves of different variables at those locations may be well mixed.

Without the spatial component, clustering itself is a well-studied problem with many established techniques such as K-means clustering (MacQueen, 1967), spectral clustering (Shi & Malik, 2000), hierarchical clustering (Nielsen, 2016), and density-based spatial clustering of applications with noise (DBSCAN, Ester et al. (1996)), to name a few popular algorithms. Each of these algorithms offers distinct advantages based on their modeling assumptions when performed on different types of data. Additionally, common extensions of these algorithms include supervised fuzzy C-means (Yasunori et al., 2009), spatial hierarchical clustering (Carvalho et al., 2009), and the generalized DBSCAN (GDBSCAN, Sander et al. (1998)) algorithm. These algorithms are able to better incorporate either response labels or spatial data directly through customized distance metrics or connectivity constraints.

However, in this paper, we consider the case of supervised spatial data, with observations consisting of three components $(s, x, y)$, where $s \in \mathbb{R}^2$ is the spatial component, $x \in \mathbb{R}^p$ is the feature component, while $y \in \mathbb{R}$ is the response variable of particular interests. Assuming that in the data there is a relationship between features $x$, or between features and geography $(s, x)$, and the response $y$, we propose a new spatial clustering algorithm based on Gaussian Processes (GPs), called Gaussian Process Spatial Clustering (GPSC), which groups together clusters based on each group's ability to predict the response variable $y$. We focus on single-output cases in this paper for simplicity, but the extension to multi-output cases where $y \in \mathbb{R}^d$ with $d > 1$ is straightforward.

For the motivating example from NC census tracts community-level data, $s$ is the longitude/latitude pairs defining each state census tract, $x$ is the set of environmental pollution variables such as levels of hexane, lead, mercury, etc, as well as average socioeconomic indicators such as unemployment rates, poverty rates, or education, and the $y$ response to be predicted is a latent class measuring socioeconomic and environmental advantage/disadvantage as defined in Larsen et al. (2020).

In order to do so, GPSC leverages the flexibility of GPs, well-studied near-universal function approximators (Wendland, 2004; Ghosal & Van der Vaart, 2017), to fit the true functional relationships within each clustering and to cluster tract locations and features pertaining to socioeconomic status. Simulation studies show that the GPSC algorithm is capable of accurately recovering and clustering these functional relationships even in cases of limited spatial dependencies and regardless of any dependencies in the covariate domain. This is important because, as in Figure 1, clusters may not always be completely separated, so it is essential to control the relative influence of each domain in the clustering done in GPSC by choosing the kernel. Furthermore, GPSC is less sensitive to dependencies in the covariate domain compared to traditional clustering methods such as K-means clustering. We prove that GPSC is able to find the true clusters as long as the functional relationships between the clusters are distinct. When applied to community-wide study, GPSC

successfully clusters tracts in NC with finer detail than traditional methods and can be interpreted by domain experts.

In summary, our contributions in this paper are 1) a novel spatial clustering GPSC algorithm, 2) theoretical support to GPSC and 3) application to NC tract level data with new interpretable discoveries. Full proofs of theorems, implementation details, as well as extended simulations are presented in the Supplementary Material.

## 2 Model

### 2.1 Gaussian Process Regression

In this section, we review the GP model and its application towards regression and classification. By definition, a GP is a random function for which any finite realization follows a multivariate Gaussian distribution Williams & Rasmussen (2006):

**Definition 2.1.** $f$ follows GP in domain $\Omega$ with mean function $\mu$ and covariance function $K$, denoted by $f \sim GP(\mu, K)$, where $\mu : \Omega \to \mathbb{R}$, $K : \Omega \times \Omega \to \mathbb{R}$, if for any $x_1, \cdots, x_n \in \Omega$,

$$[y_1, \cdots, y_n]^\top := [f(x_1), \cdots, f(x_n)]^\top \sim N(v, \Sigma),$$

where $v = [\mu(x_1), \cdots, \mu(x_n)]^\top$ and $\Sigma_{ij} = K(x_i, x_j)$.

A GP is completely determined by the mean function $\mu$ and the covariance function $K$, also known as the kernel. In this paper, we assume $\mu = 0$ for simplicity and use the radial basis function (RBF), also known as the squared exponential kernel, defined as: $K(x, x') = \sigma^2 e^{-\frac{d^2(x,x')}{2b}}$, but our model can be extended to other kernels. The two parameters, i.e., spatial variance $\sigma^2$ and length scale $b$ are estimated by maximizing the log marginal likelihood (MLE). Given training data $(x_i, y_i)_{i=1}^n$ with MLE $\theta_n = (\sigma_n^2, b_n)$ and a new observation $x_*$, the best unbiased linear predictor (BLUP, Stein (1999)) of $y_* = f(x_*)$ is given by $\widehat{y_*} = K_{\theta_n}(x_*, X)K_{\theta_n}(X, X)^{-1}Y$, where $K_{\theta_n}(x_*, X)_i = K_{\theta_n}(x_*, x_i)$, $K_{\theta_n}(X, X)_{ij} = K_{\theta_n}(x_i, x_j)$ and $Y = [y_1, \cdots, y_n]^\top \in \mathbb{R}^n$. As a flexible regression algorithm, GP can be modified into a classifier using a link function (Williams & Rasmussen, 2006) for a discrete response variable $y$. As a result, we will not distinguish between Gaussian process regression (GPR) and Gaussian process classification (GPC) in this paper.

### 2.2 GP Spatial Clustering

Now we will consider observations $\{(s_i, x_i, y_i)\}_{i=1}^n$, where $s_i \in \mathcal{S} \subset \mathbb{R}^2$ is the spatial location, $x_i \in \Omega \in \mathbb{R}^p$ is the covariate, and $y_i$ is the response variable. Let $l_i \in \{1, \cdots, L\}$ be the unobserved cluster label such that $l_i = j \iff s_i \in \mathcal{S}_j \subset \mathcal{S}$, where $\mathcal{S}_1, \cdots, \mathcal{S}_L$ is a partition of $\Omega$. We focus on the following model. $y_i = \sum_{j=1}^L \mathbf{1}_{\{s_i \in \mathcal{S}_i\}} f_j(x_i) = \sum_{j=1}^L \mathbf{1}_{\{l_i=j\}} f_j(x_i)$, where $f_j$ is unknown function on $\Omega$ in certain function class that will be discussed in Section 3. That is, the functional relation between $y_i$ and $x_i$ varies across spatial clusters supported by $\mathcal{S}_i$. The goal is to recover the cluster label $l_i$, called spatial clustering since the clusters are rooted in the spatial domain $\mathcal{S}$.

For example, in the NC tracts data, each $\mathcal{S}_i$ consists of tracts in NC, while the relationship between the latent class and the socioeconomic and environmental covariates varies across the tracts spatially. The goal is to partition NC into several clusters so that each cluster admits a unique functional relationship.

For a given observation $x_i$ in cluster $j$ with response $y_i$, we expect the prediction error of $f_j$ to be the lowest among all $f_j$'s, and hence we can assign $x_i$ to the cluster with the lowest prediction error. However, neither the cluster label $l_i$ or domain partition $\mathcal{S}_i$, nor the functions $f_j$ is observed. Motivated by the flexibility of GP models, we use GP to approximate the unobserved functions $f_j$, denoted by $\widehat{f_j}$, and assign $x_i$ to the cluster labeled by $\widehat{l_i}$ with the lowest prediction error: $\widehat{l_i} = \arg\min_j (\widehat{f_j}(s_i, x_i) - y_i)^2$. Then we update the cluster and $\widehat{f_j}$ iteratively. The GPSC algorithm is summarized in algorithm 1.

---

**Algorithm 1** Gaussian Process Spatial Clustering

---

**Input:** data $(s_i, x_i, y_i)_{i=1}^n$, number of clusters $L$, maximum number of iterations $T$
Initialize $\hat{l}_i = \text{randomInt}(1, 2, \cdots, L)$
**for** $t = 1$ **to** $T$ **do**
    **for** $j = 1$ **to** $L$ **do**
      $(S_j, X_j, Y_j) = \{(s_i, x_i, y_i) : \hat{l}_i = j\}$, $\hat{f}_j = \text{GPR}(([S_j, X_j], Y_j))$
    **end for**
    **for** $i = 1$ **to** $n$ **do**
      $\hat{l}_i = \arg\min_j (\hat{f}_j((s_i, x_i)) - y_i)^2$
    **end for**
**end for**

---

In this flexible construction, it is also possible to extend the reassignment function for different applications, such as reinforcing spatial contiguity constraints as is common in geographical clustering:

$$\hat{l}_i = \arg\min_{j=1,\cdots,L} \{(\hat{f}_j(s_i, x_i) - y_i)^2 + \lambda \|s_i - C_j\|\}$$

Here, $C_j$ is the center in the spatial domain of the current cluster $\mathcal{S}_j$, while $\lambda$ is a tuning parameter that controls the penalization of assigning points to clusters that are spatially distant. For the rest of the paper, we will focus on the case $\lambda = 0$, but will demonstrate the effects of adding such penalties in the simulation studies.

In summary, the inputs to the algorithm are observations $\{(s_i, x_i, y_i)\}_{i=1}^n$, along with tuning parameters including the number of iterations $T$ and the number of clusters $L$. In practice the number of iterations $T$ need not necessarily be large, and can be replaced with the stopping criterion when the cluster assignments stabilize. The proper choice of the number of clusters $L$ is a typical challenge in the field of clustering (Mirkin, 2011), which is beyond the scope of this paper. The choice of $L$ often requires domain expertise specific to the application at hand, see Section 5 for more detailed discussion. In practice, we also typically bound the parameters of the covariance function during optimization to prevent overfitting.

## 3 Theory

In this section, we provide theoretical support to the GPSC algorithm. We start with the necessary definitions to state the assumptions and theorems.

**Definition 3.1.** Let $K$ be a positive definite kernel on $\Omega \subset \mathbb{R}^p$, then $\mathcal{F}_K(\Omega) := \text{span}\{K(\cdot, x) : x \in \Omega\}$ with inner product form $\left(\sum_{i=1}^n a_i K(\cdot, x_i), \sum_{j=1}^m b_j K(\cdot, \widetilde{x}_j)\right)_K := \sum_{i,j} a_i b_j K(x_i, \widetilde{x}_j)$, so that $\mathcal{F}_K(\Omega)$ is a pre-Hilbert space with a reproducing kernel $K$. The linear mapping $\Phi$ $\Phi : \mathcal{F}_K(\Omega) \to C(\Omega) : \Phi(f)(x) := (f, K(\cdot, x))_K$, is injective. Then the image of $\Phi$, $\mathcal{N}_K(\Omega) := \Phi(\mathcal{F}_K(\Omega))$ is a Hilbert space with a reproducing kernel $K$ equipped with the inner product $(f, g)_K := (\Phi^{-1}f, \Phi^{-1}g)_K$.

For simplicity, we fix $K_\theta$ to be the RBF kernel with $\theta = (\sigma^2, b)$ from now on.

**Definition 3.2.** Given observations $X$ and $x_0$ with unobserved $y_0$ to be predicted. Define the following function:

$$\psi_{X,x_0} : Y \mapsto K_{\theta(Y)}(x_0, X)^\top K_{\theta(Y)}(X, X)^{-1} Y$$

where $\theta(Y) = \arg\max_\theta N(Y|0, K(X, X))$ is the maximum likelihood estimator of $\theta$ based on potential observations $Y$. That is, $\psi$ is the BLUP of $y_0 = f(x_0)$ based on observations $(X, Y)$. By the definition of $\psi$, the smoothness of the Gaussian density function and the linearity of BLUP, $\psi$ is differentiable (Stein, 1999). We also introduce the following assumptions:

(A1) $\Omega \subset \mathbb{R}^p$ is compact and $p(x) > 0$, $\forall x \in \Omega$, where $p(x)$ is the density function of $x$.

(A2) $f_j \in \mathcal{N}_K(\Omega)$, $j = 1, \cdots, L$.

**Theorem 3.3.** *Under assumptions (A1)-(A2), at any iteration in Algorithm 1, let $n_{jk} := \left| \{ i : l_i = j, \widehat{l}_i = k \} \right|$, $n_j := \left| \{ i : \widehat{l}_i = j \} \right|$ then the current $x_i$ is a assigned to the correct cluster if for any $k \neq j$,*

$$\frac{\sum_{m \neq j} n_{mj}}{\sum_{m \neq j} n_{mk}} < \frac{D_l E_l}{D_u E_u} - \frac{\|f\|_K e^{-c_1 n_j^{\frac{1}{p}}} + \|f\|_K e^{-c_2 n_k^{\frac{1}{p}}}}{D_u E_u n_{22}}, \tag{1}$$

*where $c_1$ and $c_2$ are constants, and*

$$D_l := \inf \|\nabla \psi(Y)\| \leq D_u := \|\nabla \psi(Y)\|_\infty,$$
$$E_l := \inf_{x \in \Omega, j,k=1,\cdots,L} |f_j(x) - f_k(x)|$$
$$\leq E_u := \sup_{x \in \Omega, j,k=1,\cdots,L} |f_j(x) - f_k(x)| < \infty.$$

*In particular, let $L = 2$, $j = 1$, $k = 2$ and let $n_1, n_2 \to \infty$, Equation (1) becomes: $\frac{n_{21}}{n_{22}} < \frac{D_l E_l}{D_u E_u}$. That is, the mis-clustered proportion is small enough.*

The right-hand side of inequality equation 1 is highly interpretable. The ratio $\frac{D_l}{D_u}$ measures the robustness of the BLUP, that is, how the BLUP changes with training data $Y$. The less robust the BLUP, the smaller the ratio, and the harder it is to find the correct clusters. The ratio $\frac{E_l}{E_u}$ measures the separation between functions $f_1, \cdots, f_L$. The smaller the separation, the smaller the ratio, and the harder it is to find the correct clusters. Theorem 3.3 also implies that the state of correct clustering is an absorbing state, that is, if the current clusters are close enough to the true clusters, then perfect clustering results will be achieved in the next iteration. Note that even if the inequality does not hold, the algorithm may still converge to a better state with more correctly clustered data, although not within one single step. This is because even when the right-hand side of Equation equation 1 is small, there might be some region $\Omega_0 \subset \Omega$ where the $f_j$'s are relatively well separated so that the right-hand side is relatively large on $\Omega_0$, so that samples within $\Omega_0$ will be assigned to true clusters. Meanwhile, for the region where $f_j$'s are mixed, it is challenging for all clustering algorithms.

In practice, the response variable $y$ is often subject to measurement error, leading to a more realistic model: $y = f(x) + \epsilon$, where $\epsilon \sim N(0, \tau^2)$ represents noise. The following theorem serves as the counterpart to Theorem 3.3 in the presence of Gaussian noise:

**Theorem 3.4.** *Under the same assumption and notation as of Theorem 3.3, with the addition of Gaussian noise, the current $x_i$ is assigned to the correct cluster if for any $k \neq j$,*

$$\frac{\sum_{m \neq j} n_{mj}}{\sum_{m \neq j} n_{mk}} < \frac{D_l E_l}{D_u E_u} - \frac{\|f\|_K e^{-c_1 n_j^{\frac{1}{p}}} + \|f\|_K e^{-c_2 n_k^{\frac{1}{p}}} + \xi}{D_u E_u n_{22}}, \tag{2}$$

*where $\xi$ is the sum of independent $\chi$-distributions with degrees of freedom $1, n_1$ and $n_2$ rescaled by $2\tau$, $\tau$ and $\tau$ respectively.*

*In particular, when $L = 2$, $j = 1$, $k = 2$, and $n_1, n_2 \to \infty$, the right-hand side simplifies to $\frac{D_l E_l}{D_u E_u}$ with probability one.*

When $\tau = 0$, that is, the noise vanishes, then $\xi = 0$ so Theorem 3.4 coincides with Theorem 3.3.

## 4  Simulation Studies

To evaluate the performance of GPSC, we present three simulation studies in this section, with detailed implementation details in the Supplementary Materials. The first simulation will demonstrate an application of algorithm 1 in the case of responses generated by linear functions with two clusters, while the second

simulation shows the performance of GPSC in the case of responses generated by nonlinear functions. The third simulation shows the robustness of GPSC to noisy data and overspecified number of clusters. In all simulations, we compare the performance of GPSC with traditional clustering algorithms: K-means, spectral clustering, hierarchical clustering, and DBSCAN, as well as spatial or supervised analogs: supervised fuzzy C-means, spatial hierarchical clustering, generalized GDBSCAN, and also the Gaussian mixture model (GMM, Day (1969)). We evaluate the performance using the adjusted Rand index (ARI, Steinley (2004)) and adjusted mutual information (AMI, (Strehl & Ghosh, 2002)) against the true labels. The data used in these simulations take the form $\{(s_i, x_i, y_i)\}_{i=1}^n$, where $s_i \in \mathbb{R}^2$ is the spatial domain, $x_i \in \mathbb{R}^2$ is the covariate domain, and $y_i \in \mathbb{R}$ is the response domain, taken for visualization purposes. Note that for all algorithms, including GPSC and the aforementioned traditional, nonspatial clustering algorithms, the input is taken to be the full vector $(s, x, y)$ with the spatial domain included, so that all competitors always use the full information. The results can be directly extended to higher $p$ and multivariate responses.

### 4.1 Simulation 1 - Linear Functions

In this simulation, $y$ is a linear function of $x$ for visualization purposes, where both $s_i$ and $x_i$ are generated from independent uniform distributions. After generating the data $\{(s_i, x_i)\}_{i=1}^n$, the spatial domain is subdivided into two clusters, the center ball and the background region. The $y_i \in \mathbb{R}$ are then generated as distinct linear functions of $x_i$ for each cluster. For visualizations of the resulting clusters in the $XY$ domain and all ARI/AMI scores, see Supplement D.1.

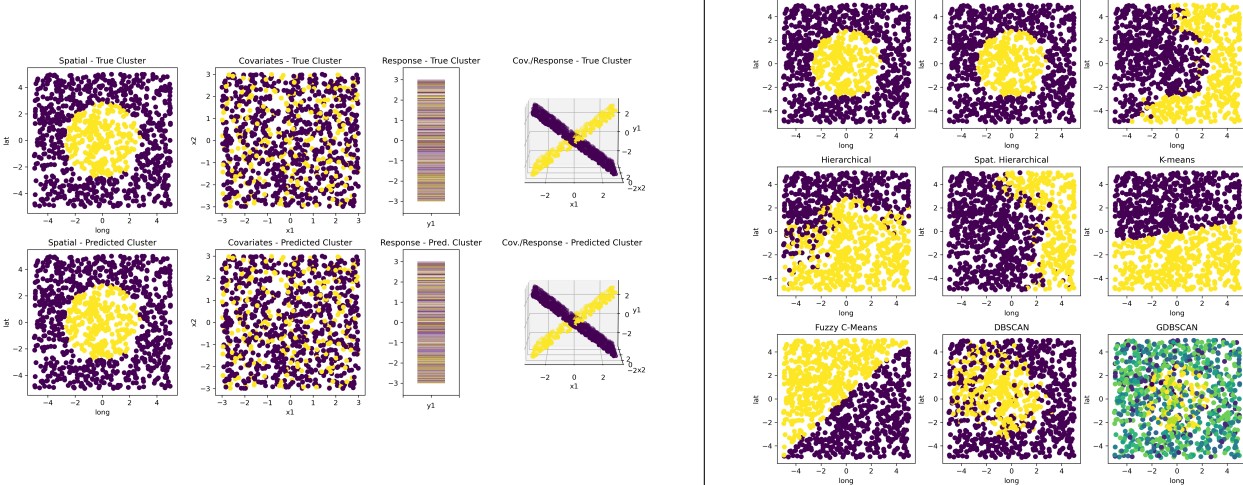

Figure 2: [Left] GPSC results for Simulation 1, colored by cluster. The first column plots the spatial domain $s_i \in \mathbb{R}^2$, the second column plots the covariate space $x_i \in \mathbb{R}^2$, the third column plots the response space $y_i \in \mathbb{R}$ , while the right-most column plots $y_i \in \mathbb{R}$ against $x_i \in \mathbb{R}^2$. The first row shows the ground truth generated data. The second row shows the predicted clusters from GPSC after randomized initialization. [Right] Clusters for Simulation 1 by nine clustering algorithms visualized in the spatial domain.

It can be seen that this simulation is challenging for several reasons. First, there is almost no separation considering any dimension $s$, $x$, or $y$ on its own as in the first three columns in Figure 2 (left); the separation is solely in the functional domain $XY$. As a result, most traditional algorithms cannot capture this functional relationship, as supported by Panels 3-7 in Figure 2 (right). Although it can seen that the Gaussian mixture model is able to rediscover the clusters in this case (Panel 2), this is due to GMM's ability to estimate the pairwise linear correlation between each domain. However, we expect GMM to fail to capture nonlinear functional relationships, as shown in the following Simulation 2. It is also noted that DBSCAN and GDBSCAN (Panels 8 and 9) also perform reasonably well, but have challenges of their own such as GDBSCAN greatly overestimating the number of clusters.

## 4.2 Simulation 2 - Nonlinear Functions

In this simulation, we will show that in an irregular spatial distribution with nonlinear relationships between the covariates and the response variable, GPSC is still able to recover the true functional relationships in contrast to the competitors. After generating the data $\{(s_i, x_i)\}_{i=1}^n$ from independent uniform distributions, the spatial domain is subdivided into two clusters, the ring and the background region. The $y_i \in \mathbb{R}$ are then generated as distinct nonlinear functions of $x_i$ for each cluster (the first row of Figure 3).

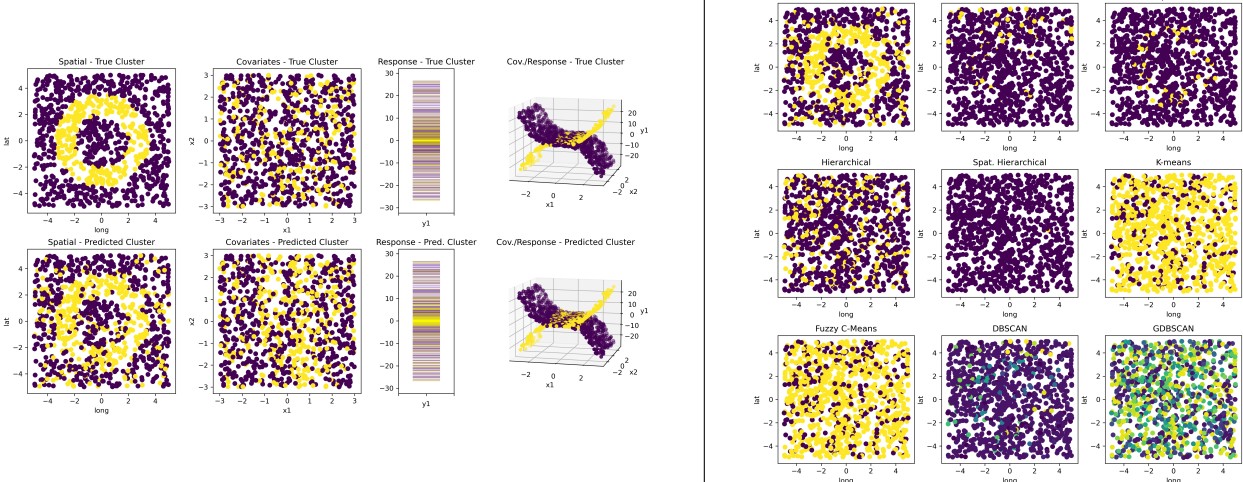

Figure 3: [Left] Results for Simulation 2 with true generated data (top) and results of GPSC (bottom). [Right] Clusters for Simulation 2 by nine clustering algorithms visualized in the spatial domain.

It can be seen that in this more challenging simulation, only GPSC is able to recover the true functional clusters, with the results of each clustering algorithm plotted in the spatial domain in Figure 3 (see Supplement D.2 for more details).

## 4.3 Simulation 3 - Model Robustness

In Simulation 3, we present a more realistic scenario of three clusters that have some degree of spatial separation. Motivated by our real-world application of clustering North Carolina census tracts, the sun and moon clusters could be interpreted to represent two urban centers surrounded by a larger rural region. By applying the spatially penalized version of GPSC, we will show that the clustering results remain stable across both increasing levels of noise, as well as to overspecification of the input number of clusters. Full visualization and comparisons can be found in Supplement D.3, D.4 and D.5.

After generating the data $\{(s_i, x_i)\}_{i=1}^n$ from independent uniform distributions, the spatial domain is sub-divided into the three clusters, the sun and moon shape, and the background region. The $y_i \in \mathbb{R}$ are then generated as distinct nonlinear functions of $x_i$ for each cluster with varying degrees of zero-mean Gaussian noise. For an extension of Simulation 3 to nonlinear functions of both $s_i$ and $x_i$, see Supplement D.5.

### 4.3.1 Noisy Responses

In this section, we show that GPSC works under noisy conditions as per Theorem 3.4. In Figure 25, we present Simulation 3 with noise variance = 100, showing that the spatially penalized version of GPSC still performs well under noisy conditions. In particular, GPSC is able to outperform competitors at all tested noise levels, where no other competitor is able to recover the true clusters (with exact ARI/AMI scores and additional details in Supplement D.3).

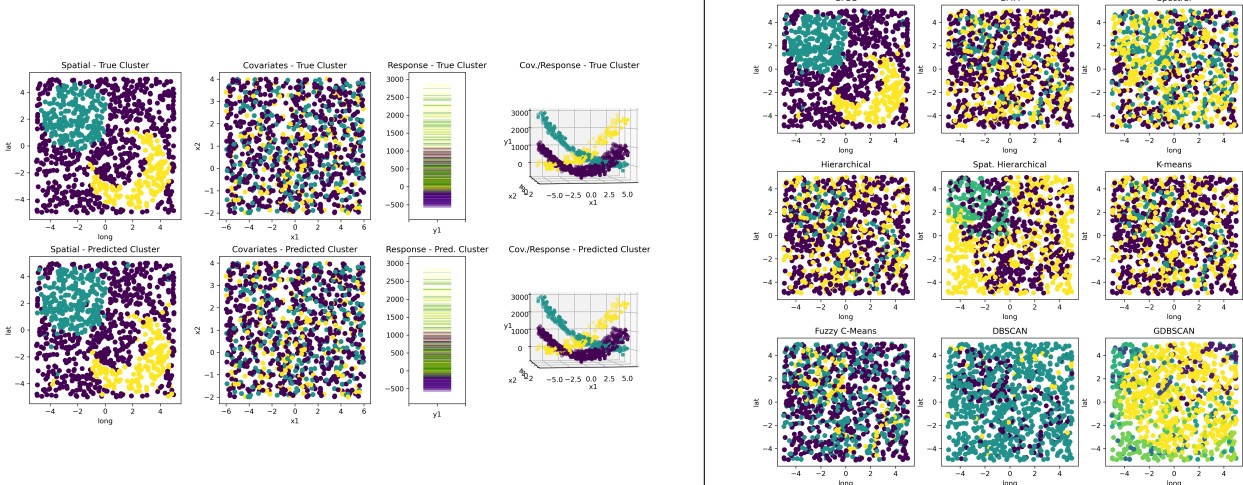

Figure 4: [Left] Results for Simulation 3 with true generated data (top) and results of GPSC (bottom). [Right] Clusters for Simulation 3, by nine clustering algorithms visualized in the spatial domain.

### 4.3.2 Overspecified Number of Clusters

Finally, we show that GPSC is stable when the number of clusters is overspecified. Specifically, it can be seen in Figure 5 when the number of specified clusters is 5, the sun (teal) and moon (yellow) clusters remain stable, while the background cluster (originally purple) is split into three purple, indigo, and light green clusters. In contrast, the competitors are unable to recover the true clusters when the number of clusters are overspecified, while further visualizations and comparisons to the competitor models are presented in Supplement D.4.

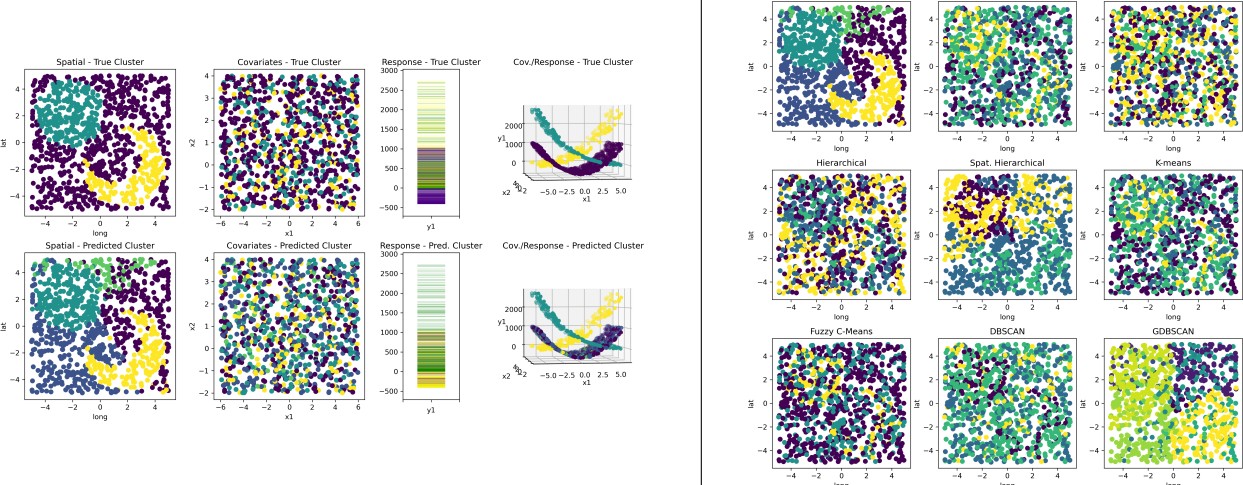

Figure 5: [Left] GPSC results for Simulation 3 with overspecified number of clusters as 5, along with competitors. [Right] Results of nine algorithms with overspecified input presented in the spatial domain.

## 5 Applications to NC Tract Data

This dataset consists of 29 community-wide covariates aggregated by census tracts in North Carolina. Such covariates ranged from measures of environmental pollution to averages of socioeconomic indicators such as unemployment, housing environment, education, etc (see Supplement E for a full list). Each census tract is

associated with a single (longitude, latitude) pair of coordinates. The overall socioeconomic indicators were previously aggregated using latent class analysis into a single advantage/disadvantage class ranging from 1-8 Larsen et al. (2020).

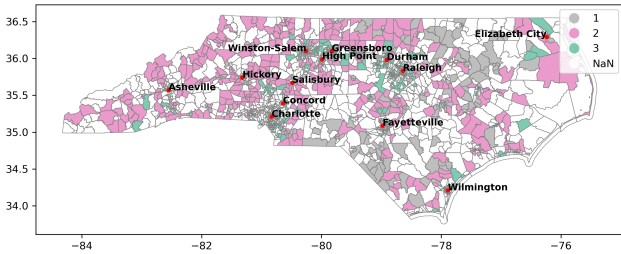

Figure 6: Baseline aggregate groups of socioeconomic and environmental latent class indicator.

Based on the distribution of the full latent classes seen in Figure 1, we can see that there is some degree of separation in the spatial domain between certain groups. Thus, we initialized our GPSC algorithm by performing traditional K-means clustering on solely the spatial domain. We then applied our GPSC algorithm using this latent class as the response variable, taking all other features as the set of covariates, and compared the results with K-means clustering for comparison. Here, we focus on K-means for comparison due to its interpretable results from previous studies in Larsen et al. (2020), with results from other algorithms presented in Supplement E. Based on our results, we find that $L = 3$ produced the most interpretable clusters, and thus aggregated the 8 latent classes into 3 to visualize as a baseline against GPSC seen in Figure 6. Using the language of Larsen et al. (2020) for our predicted 3 clusters, we will consider the overall socioeconomic and environmental advantage to be three levels: low (pink), medium (gray), and high (green).

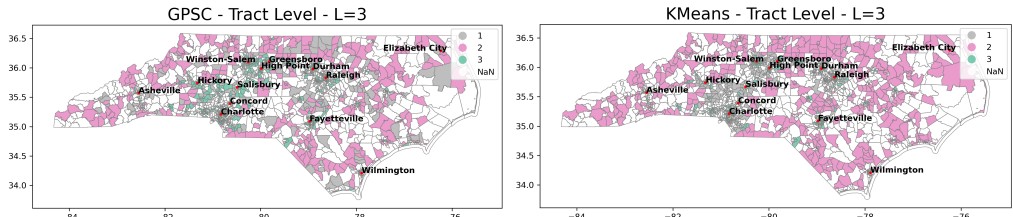

Figure 7: Clusters by GPSC and K-means for tract data, interpreted as overall socioeconomic/envrional advantage between levels of low (pink), medium (grey), and high (green).

At first glance, the general spatial distribution of our GPSC and K-means algorithms tends to agree. However, the GPSC predicted clusters differ from K-means and baseline in several meaningful ways. First, in the central region depicted in the first row of Figure 8, GPSC identifies more areas of high advantage (green). Notably, this includes the area surrounding cities such as Chapel Hill, Cary, and the capital city Raleigh (Research Triangle Park), as well as Greensboro and High Point (the Piedmont Triad), which are known to be wealthier and more urbanized regions of the state, whereas the K-means algorithm puts tracts within this region in the medium (gray) advantage group.

Towards the edges of the state we can also see significant differences as the GPSC algorithm tends to further differentiate tracts around the extremities between low and medium advantage. Most notably, around Asheville and Wilmington, two more prominent cities in North Carolina, we are able to distinguish further differences between low and medium advantage tracts, as seen in the second and third rows in Figures 8. Considering the ARI and AMI scores between the two clusterings, we find the scores to be both 0.002, suggesting that clusterings, despite visually seeming to separate the tracts spatially in similar patterns, are actually very different. One challenge of K-means clustering in Larsen et al. (2020) when determining the original 8 latent classes was a potential lack of finer detail from the K-means predicted clusters. However, here we have shown that despite using the same $L = 3$ clusters, GPSC is able to further differentiate between areas of low and medium disadvantage, in less dense areas of the state along the coast and the western region.

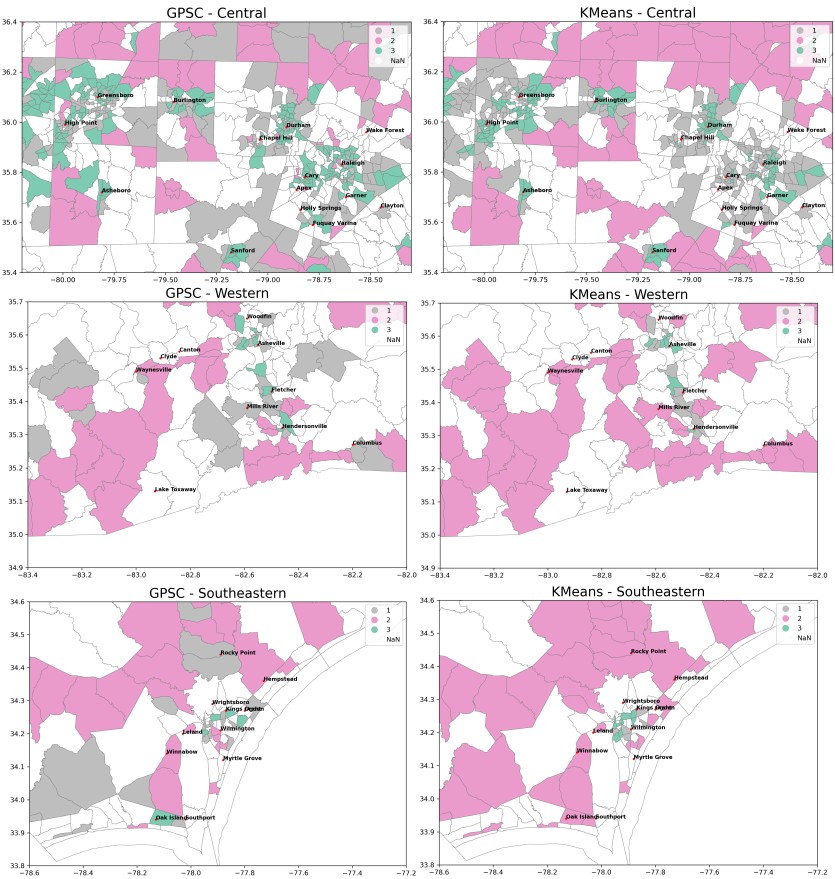

Figure 8: GPSC and K-means cluster results for NC tracts. Row 1: Central NC; Row 2: Western NC (Asheville); Row 3: Southeastern NC (Wilmington)

Furthermore, there is reason to believe that not all 8 classes are necessary to describe the different advantage groups. In the original grouping, the latent class 2 is actually an empty group, as seen in Figure 1. Thus, the results from GPSC in comparison to K-means and baseline suggest that the algorithm is able to better balance nuance against a traditional clustering algorithm, while also retaining simpler interpretability by using fewer clusters.

## 6  Discussion

Spatial clustering offers unique challenges in comparison to traditional clustering problems due to the spatial domain inherent to geographic data. In our application, the census tract data have distinctly different properties compared to the measured covariates over the tracts. In this paper, we propose a GP-based clustering algorithm and demonstrate its performance in both simulation studies and a real data application. The advantages of GPSC include being able to capture the relative effects between the spatial domain and the measured covariates, largely independent of intersections in the covariate domain as long as the clustered functions themselves have some degree of separation. We also provide theoretical guarantees to the convergence of GPSC and extend it to noisy settings. In the simulations, we demonstrate these scenarios in which the clusters were mostly inseparable when considering any single domain, yet the GPSC model outperforms all competitors in recovering the true cluster by fitting the relationship between the covariate and the response.

GPSC can also be highly scalable; the complexity of the algorithm stems from the fitting of each GP in each iteration, where standard Gaussian processes regression is $O(n^3)$ in the size of the input. In our case,

we applied a standard Gaussian process regression model from the scikit-learn package (Pedregosa et al., 2011) since our sample size was relatively small. However, in cases of large sample size, scalable GP methods can be applied for a reduction in runtime to $O(n \log n)$ (Liu et al., 2020). The GPSC model also has few tuning parameters, notably the number of clusters, optional spatial penalty for data thought to contain spatially contiguous clusters, and and can also be highly flexible through the choice of GP kernel. Although the form of our theorem is independent of the specific choice of kernel (only the convergence rate will differ), in practice more nuanced anisotropic or nonstationary kernels may be more suitable for datasets with strong heterogeneity, for which the actual design of such kernels remains an open problem.

In the real-world application, we applied GPSC to a North Carolina socioeconomic and environmental indicator dataset and found distinct patterns of advantage-disadvantage across the state that captured finer details around the less dense outer regions of the state in comparison to K-means and other clustering methods (presented in Supplement E), while our method also offered simpler interpretability than previous analysis. When utilized by domain experts, the goal of the results of these models is to supplement the identification of marginalized communities, which could be targeted with interventions.

### 6.1 Broader Impacts Statement

In context of our long-term goal of designing interventions, ensuring the accuracy of these models is also of high ethical importance. Therefore in our case, before any application, we can perform sensitivity analyses that tile the geographic region with alternative regional classifiers (county, AHEC region, latitude and longitude tiles of uniform size) to confirm that the same areas arise in multiple boundary definitions. This will confirm that the boundary definitions are not driving artifactual associations. More broadly, it is important that in these high-stakes applications we do not over-rely on any one method. We envisage the possibility of using these clustering results (and GPSC in general) as a supplementary tool for experts to potentially better identify marginalized communities and areas that may be otherwise overlooked.

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

## A    Appendix Overview

In Section B we continue the discussion section of the main paper on the potential limitations of our algorithm and provide some general guidance on usage. In Section C we provide proofs for the two convergence theorems presented in the main paper. Section D contains the full implementation details, and additional figures for the three simulation studies presented in the main paper, as well as an extension to spatially-dependent functions. Section E contains additional details for the real-world application of the main paper as well as additional comparisons to competitor algorithms. All code is available anonymously at https://anonymous.4open.science/r/gpsc-tmlr24-CD3D/README.md.

# B    Potential Limitations

Here we continue our discussion from the paper on potential limitations of the GPSC model. The first is the question of tuning the appropriate number of clusters. This is a well-known challenge in clustering, which is beyond the scope of our study. However, some clustering algorithms have the capability to automatically determine the number of clusters. In this regard, we found that DBSCAN and GDBSCAN generally performed poorly in our simulation studies, resulting in incorrect and irrelevant clusters. As such, like in our real world application, selecting the appropriate number of clusters for the problem is best handled on a case-by-case basis with input from domain experts or prior background knowledge. The same can be said of the optional tuning parameter $\lambda$, which again reinforces contiguous spatial restraints by penalizing assignments to distant clusters. In our application, we were able to compare our results with previous studies on the socioeconomic and environmental cancer risk factors across the state, as well as collaborate with epidemiologists and cancer experts familiar with the datasets.

Next we briefly revisit the modeling assumptions of GPSC. Our main focus is spatial clustering, as motivated by the CBCS application, where different spatial clusters exhibit different functional relations between the response variable y and input features. In this case, according to our theorems presented in the main paper, the performance of GPSC is influenced by several key factors, including $D_u$, $D_l$, $E_u$, and $E_l$. In simpler terms, if the true underlying functions $f_j$ in different clusters are not clearly distinguishable, or they have unbounded derivatives, it may be challenging to achieve optimal clustering results. However, this is a common challenge for most clustering algorithms, and overcoming this limitation may require more advanced techniques and designs.

# C    Proofs

## C.1    Proof of Theorem 3.3

We first consider the case of $L = 2$ that is, there are two clusters. Let $l_i$ be the unobserved true cluster label of the $x_i$ and $\widehat{l}_i$ be the cluster label of $x_i$ in the current iteration. Let $x_0$ be a sample to be clustered with (unobserved) label $l_0 = 1$, that is, $x_0$ should be assigned to cluster-1. Our goal is to show that GPSC does assign $x_0$ to cluster-1 under the condition explicitly stated in Theorem 3.3.

Let $(X_1, Y_1) \coloneqq \{(x_i, y_i) : \widehat{l}_i = 1\}$ be the set of samples assigned to cluster-1 with size $n_1 \coloneqq \#\{i : \widehat{l}_i = 1\}$. Similarly, let $(X_2, Y_2) \coloneqq \{(x_i, y_i) : \widehat{l}_i = 2\}$ be the set of samples assigned to cluster-2 with size $n_2 \coloneqq \#\{i : \widehat{l}_i = 2\}$. According to Algorithm 1, we train two GPR models based on $(X_1, Y_1)$ and $(X_2, Y_2)$, to obtain two predictors of $y_0$ denoted by $\widehat{y}^{(1)}$ and $\widehat{y}^{(2)}$. Under the notation in Definition 3.2, we have $\widehat{y}^{(1)} = \psi_{X,x_0}(Y_1)$ and $\widehat{y}^{(2)} = \psi_{X,x_0}(Y_2)$, it suffices to show that $e_1 \coloneqq |y_0 - \widehat{y}^{(1)}| < e_2 \coloneqq |y_0 - \widehat{y}^{(2)}|$ as long as

$$\frac{n_{21}}{n_{22}} < \frac{D_l E_l}{D_u E_u} - \frac{\|f\| e^{-c_1 n_1^{\frac{1}{p}}} + \|f\| e^{-c_2 n_2^{\frac{1}{p}}}}{D_u E_u}.$$

To calculate $e_1$, we introduce the following partially observed dummy variables $\widetilde{Y}_1 \coloneqq f_1(X_1)$ and let $\widetilde{y}^{(1)} \coloneqq \psi_{X_1,x_0}(\widetilde{Y}_1)$. We plug this term in $e_1$ and apply triangle inequality to obtain the following:

$$e_1 = |y_0 - \widehat{y}^{(1)}| = |y_0 - \widetilde{y}^{(1)} + \widetilde{y}^{(1)} - \widehat{y}^{(1)}| \leq \underbrace{|y_0 - \widetilde{y}^{(1)}|}_{\text{①}} + \underbrace{|\widetilde{y}^{(1)} - \widehat{y}^{(1)}|}_{\text{②}}.$$

Observe that ① is the prediction error of standard Gaussian process regression on $(X_1, \widetilde{Y}_1)$, without any misspecified samples. As a result, the upper bound of ① comes from Lemma C.1, the asymptotic theory of Gaussian process regression. That is, ① $\leq \|f\| e^{-c_1/h_{n_1}}$ for some constant $c_1$. Assumption (A1) and Dudley's theorem  imply that $h_{n_1} = O(n_1^{-\frac{1}{p}})$, so ① $\leq \|f\| e^{-c_1 n_1^{\frac{1}{p}}}$.

To analyze ②, we first observe that $\widehat{y}^{(1)} = \psi_{X_1,x_0}(Y_1)$ is based on partially correct clusters, while $\widetilde{y}^{(1)} = \psi_{X_1,x_0}(\widetilde{Y}_1)$ is based on true clusters. Then by the differentiability of $\psi$, we have

$$② = |\psi_{X_1,x_0}(Y_1) - \psi_{X_1,x_0}(\widetilde{Y}_1)| \leq \|\nabla \psi_{X_1,x_0}\|_\infty \|Y_1 - \widetilde{Y}_1\| = D_u \|Y_1 - \widetilde{Y}_1\|,$$

where $D_u = \sup_{X_1 \subset X, x_0 \in X} \|\nabla \psi_{X_1,x_0}\|_\infty$. As a result, it suffices to find an upper bound of $\|Y_1 - \widetilde{Y}_1\|$.

Observe that among samples in $(X_1, Y_1)$, some are correctly clustered, denoted by $(X_{11}, Y_{11}) = \{(x_i, y_i) : l_i = 1, \widehat{l}_i = 1\}$ with size $n_{11}$, while the rest are incorrectly clustered, denoted by $(X_{21}, Y_{21}) = \{(x_i, y_i) : l_i = 2, \widehat{l}_i = 1\}$ with size $n_{21}$. After reordering the samples, we have $X_1 = \begin{bmatrix} X_{11} \\ X_{21} \end{bmatrix}$ and $Y_1 = \begin{bmatrix} Y_{11} \\ Y_{21} \end{bmatrix}$. By the model assumption, for the correctly clustered samples $Y_{11} = f_1(X_{11})$, while for the incorrectly clustered samples, $Y_{21} = f_2(X_{21}) \neq f_1(X_{21})$. By the same rule, we can split $\widetilde{Y}_1$ into two components as well, i.e., $\widetilde{Y}_1 = \begin{bmatrix} \widetilde{Y}_{11} \\ \widetilde{Y}_{21} \end{bmatrix}$ with $\widetilde{Y}_{11} = f_1(X_{11}) = Y_{11}$ and $\widetilde{Y}_{21} = f_1(X_{21})$. That is, the difference between $Y_1$ and $\widetilde{Y}$ only comes from $Y_{21}$ and $\widetilde{Y}_{21}$:

$$\begin{aligned} |Y_1 - \widetilde{Y}_1\| &= \left\| \begin{bmatrix} Y_{11} \\ Y_{21} \end{bmatrix} - \begin{bmatrix} \widetilde{Y}_{11} \\ \widetilde{Y}_{21} \end{bmatrix} \right\| = \left\| \begin{bmatrix} f_1(X_{11}) \\ f_2(X_{21}) \end{bmatrix} - \begin{bmatrix} f_1(X_{11}) \\ f_1(X_{21}) \end{bmatrix} \right\| \\ &= \|f_2(X_{21}) - f_1(X_{21})\| \leq n_{21}\|f_2 - f_1\|_\infty = n_{21} E_u, \end{aligned}$$

where $E_u = \|f_2 - f_1\|_\infty$. Combining ① and ②, we derive the upper bound of $e_1$:

$$e_1 \leq C_1 e^{-c_1 n_1^{\frac{1}{p}}} + n_{21} D_u E_u.$$

Then we calculate $e_2$ by similar idea, but with all inequalities reversed. Again, we introduce the partially unobserved variables $\widetilde{Y}_2 := f_1(X_2)$ and let $\widetilde{y}^{(2)} := \psi_{X_2,x_0}(\widetilde{Y}_2)$. Again, by triangle inequality, we fin the following lower bound of $e_2$:

$$e_2 = |y_0 - \widehat{y}^{(2)}| = |y_0 - \widetilde{y}^{(2)} + \widetilde{y}^{(2)} - \widehat{y}^{(2)}| \geq \underbrace{|\widetilde{y}^{(2)} - \widehat{y}^{(2)}|}_{③} - \underbrace{|y_0 - \widetilde{y}^{(2)}|}_{④}$$

Finding the upper bound for ④ follows similar logic as for the upper bound for ①. Observe that ④ is the prediction error of standard Gaussian process regression on $(X_2, \widetilde{Y}_2)$, without any misspecified samples. As a result, the upper bound of ④ comes from Lemma C.1 and Assumption (A1). That is, $④ \leq \|f\| e^{-c_2 n_2^{\frac{1}{p}}}$ for some constant $c_2$.

While, unlike finding upper bound for ②, our goal is to find a lower bound for ③. By mean value theorem,

$$③ = |\psi_{X_2,x_0}(\widetilde{Y}_2) - \psi_{X_2,x_0}(Y_2)| \geq \inf \|\nabla \psi_{X_2,x_0}(Y)\|_\infty \|Y_1 - \widetilde{Y}_1\| = D_l \|\widetilde{Y}_2 - Y_2\|.$$

To find the lower bound of $\|\widetilde{Y}_2 - Y_2\|$, we again split both vectors into two components: $X_2 = \begin{bmatrix} X_{12} \\ X_{22} \end{bmatrix}$ and $Y_2 = \begin{bmatrix} Y_{12} \\ Y_{22} \end{bmatrix}$, where $Y_{12} = f_2(X_{12})$ and $Y_{22} = f_1(X_{22})$. Then,

$$\begin{aligned} \|Y_2 - \widetilde{Y}_2\| &= \left\| \begin{bmatrix} Y_{12} \\ Y_{22} \end{bmatrix} - \begin{bmatrix} \widetilde{Y}_{12} \\ \widetilde{Y}_{22} \end{bmatrix} \right\| = \left\| \begin{bmatrix} f_1(X_{12}) \\ f_2(X_{22}) \end{bmatrix} - \begin{bmatrix} f_1(X_{12}) \\ f_1(X_{22}) \end{bmatrix} \right\| \\ &= \|f_2(X_{22}) - f_1(X_{22})\| \geq n_{22} \inf_{x \in \Omega} |f_2(x) - f_1(x)| \geq n_{22} E_l. \end{aligned}$$

Combining ③ and ④, we find the lower bound of $e_2$:

$$e_2 \geq n_{22} D_l E_l - C_2 e^{-c_2 n_2^{\frac{1}{p}}}.$$

Finally, we conclude that $e_1 < e_2$ if $C_1 e^{-c_1 n_1^{\frac{1}{p}}} + n_{21} D_u E_u < n_{22} D_l E_l - C_2 e^{-c_2 n_2^{\frac{1}{p}}}$, that is inequality 1.

**Lemma C.1** ([Wendland (2004)](#)). *When $f \in \mathcal{N}_K(\Omega)$ where $K$ is the RBF kernel in $\mathbb{R}^p$, then let $\widehat{f}_n$ be the approximation to $f$ by GP based on training samples $(X, Y)$ with sample size $n$ and filled distance $h_n \coloneqq \sup_{x \in \Omega} \min_i \|x - x_i\|$, then*

$$\|f - \widehat{f}_n\|_\infty \le e^{-c/h_n} \|f\|_K. \tag{3}$$

To prove Theorem 3.3 for arbitrary $L$ and $j$, the only difference is in the construction of $\widetilde{Y}_j$, which splits into $L$ components. To analyze $e_j$, $\widetilde{Y}_j = [\widetilde{Y}_{1j}, \cdots, \widetilde{Y}_{Lj}]^\top$ where $\widetilde{Y}_{kj} = f_j(Y_{kj})$ with $\widetilde{Y}_{jj} = Y_{jj}$. As a result, $\|\widetilde{Y}_j - Y_j\| \le \sum_{k \ne j} n_{kj} D_u E_u$ and

$$e_j \le \|f\| e^{-c_1 n_j^{\frac{1}{p}}} + \sum_{k \ne j} n_{kj} D_u E_u.$$

Similarly, for $e_k$ with $k \ne j$, $\widetilde{Y}_k = [\widetilde{Y}_{1k}, \cdots, \widetilde{Y}_{Lk}]^\top$ where $\widetilde{Y}_{mk} = f_j(Y_{mk})$ with $\widetilde{Y}_{jk} = Y_{jk}$. As a result, $\|\widetilde{Y}_k - Y_k\| \ge \sum_{m \ne k} n_{mk} D_l E_l$ and

$$e_k \ge \sum_{m \ne k} n_{kj} D_l E_l - \|f\| e^{-c_k n_k^{\frac{1}{p}}}.$$

## C.2   Proof of Theorem 3.4

For simplicity, we show the case of $L = 2$ only, the extension to general case is similar to the proof of Theorem 3.3. Following the proof in Section C.1, it suffices to analyze ①. Recall that $y_0 = f_1(x_0) + \epsilon$, we first define $y_{0*} \coloneqq f_1(x_0)$, then $|y_0 - y_{0*}| \le |\epsilon| \le 3\tau$ with probability 99.7% since $\epsilon \sim N(0, \tau^2)$. Since $y_{0*}$ is the clean observation without any noise, the previous analysis carries to $|y_{0*} - \widetilde{y}(1)|$ naturally, that is,

$$① = |y_0 - \widetilde{y}^{(1)}| = |y_0 - y_{0*} - y_{0*} + \widetilde{y}^{(1)}|$$

$$\le |y_0 - y_{0*}| + |y_{0*} - \widetilde{y}^{(1)}| \le |\epsilon_0| + \|f\| e^{-c_1 n_1^{\frac{1}{p}}}$$

where $\epsilon_0 \sim (0, \tau^2)$. To analyze ②, by the same argument, it suffices to bound $\|Y_1 - \widetilde{Y}_1\|$.

$$\|Y_1 - \widetilde{Y}_1\| = \left\| \begin{bmatrix} Y_{11} \\ Y_{21} \end{bmatrix} - \begin{bmatrix} \widetilde{Y}_{11} \\ \widetilde{Y}_{21} \end{bmatrix} \right\| = \left\| \begin{bmatrix} f_1(X_{11}) \\ f_2(X_{21}) \end{bmatrix} + \Delta - \begin{bmatrix} f_1(X_{11}) \\ f_1(X_{21}) \end{bmatrix} \right\|$$

$$= \|f_2(X_{21}) - f_1(X_{21})\| + \|\Delta_1\| \le n_{21} \|f_2 - f_1\|_\infty = n_{21} E_u + \|\Delta_1\|,$$

where $\Delta_1$ is the vector of noise $\epsilon$'s so $\|\Delta_1\| \sim \chi(n_1)$. As a result,

$$e_1 \le C_1 e^{-n_1^{\frac{1}{p}}} + n_{21} D_u E_u + |\epsilon_0| + \|\Delta_1\|.$$

In the same logic, we have $③ \ge n_{22} D_l E_l - \|\Delta_2\|$, $④ \le |\epsilon| + \|f\| e^{-c_2 n_2^{-\frac{1}{p}}}$, and

$$e_2 \ge n_{22} D_l E_l - C_2 e^{-n_2^{\frac{1}{p}}} - |\epsilon_0| - \|\Delta_2\|.$$

Finally, we conclude that $e_1 < e_2$ if

$$C_1 e^{-n_1^{\frac{1}{p}}} + n_{21} D_u E_u + |\epsilon_0| + \|\Delta_1\| < n_{22} D_l E_l - C_2 e^{-n_2^{\frac{1}{p}}} - |\epsilon_0| - \|\Delta_2\|,$$

that is,

$$n_{21} D_u E_u < n_{22} D_l E_l - C_1 e^{-c_1 n_1^{\frac{1}{p}}} - C_2 e^{-c_2 n_2^{\frac{1}{p}}} - 2|\epsilon| - \|\Delta_1\| - \|\Delta_2\|.$$

Note that $2|\epsilon| = 2\tau \chi(1)$, $\|\Delta_1\| = \tau \chi(n_1)$ and $\|\Delta_2\| = \tau \chi(n_2)$. Then Theorem 3.4 follows by setting $\xi = 2|\epsilon| + \|\Delta_1\| + \|\Delta_2\|$, the sum of independent $\chi$-distributions with degrees of freedom $1, n_1$ and $n_2$ rescaled by $2\tau, \tau$ and $\tau$ respectively.

The limiting case holds since $\chi(n)/n \xrightarrow{n \to \infty}$, that is, the $\chi$ random variable grows sub-linearly with the degree of freedom.

# D   Details on Simulation Studies

All simulation experiments were carried out on an Apple Macbook Pro with M1 Pro processor with 32 GB of memory. The scikit-learn clustering package Pedregosa et al. (2011) and scikit-fuzzy Warner & Sexauer package were used for all experiments to perform traditional clustering as well as handling Gaussian process regression for the GPSC algorithm and computing clustering metrics. All code for the simulation studies has also been made available. Note that for all simulations in the main paper, as well as in each of the additional simulations presented in this section, GPSC and all other competitor algorithms were tuned on a single random seed. Experiments were then replicated using these same parameters 49 more times on the next 49 random seeds in order for a total of 50 replicates, and the adjusted Rand index and adjusted mutual information scores are reported as mean $\pm$ standard deviation. Finally, an early stopping condition was employed for all experiments. When both the adjusted Rand index and adjusted mutual information both are above 0.90 (exact value can be set by user) on an iteration by iteration basis, the algorithm is thought to have converged and stopped. The exact values for all experiments are presented with the code.

For parameter tuning of the competitor methods, a grid search over the parameters maximizing the adjusted mutual information score against the true labels was performed as followed: 1) For K-means, the default parameters were used in the scikit-learn package. 2) For spectral clustering, the affinity matrix was determined by nearest neighbors, where the number of neighbors was tuned between 1 and 50 in increments of one. 3) For DBSCAN, the eps parameter (maximum distance between two samples in a single neighborhood) was searched between 1 and 100 in increments of one, and the minimum number of samples in a neighborhood was tuned between 1 and 40 in increments of one. 4) For standard hierarchical clustering, the default parameters were used under the ward linkage. 5) For supervised fuzzy C-means, the default arguments in the scikit-fuzzy package were used, except the algorithm was initialized using the response variable $y$ as the labels. 6) For GDBSCAN, the distance thresholds were tuned individually for each simulation. 7) For spatialized hierarchical clustering, the spatial connectivity matrix was determined by k-nearest neighbors, where the number of neighbors was searched between 1 and at least 75 in increments of one, and where the linkage was also varied between the set {average, complete, ward, single}. 8) Finally for the Gaussian mixture model, the default parameters in the scikit-learn package were used. Any auxiliary parameters unspecified here were left as the default values from the packages.

## D.1   Simulation 1

The data used in this simulation takes the form $\{(s_i, x_i, y_i)\}_{i=1}^n$, where $s_i \in \mathbb{R}^2$, the spatial domain, $x_i \in \mathbb{R}^2$, the covariate domain, and $y_i \in \mathbb{R}$, the response domain, for visualization purposes.

In this simulation, both $s_i \in \mathbb{R}^2$ and $x_i \in \mathbb{R}^2$ are generated from independent uniform distributions, where $s_i \sim \text{Unif}(-5, 5)$ and $x_i \sim \text{Unif}(-3, 3)$ component-wise.

After generating the data $\{(s_i, x_i)\}_{i=1}^n$, where $n = 1000$ samples, the domain square is subdivided into two clusters, the ball shape cluster and the rest region. This is done by subsetting all points $\{(s_i, x_i)\}$ within 2.8 units of the point $(0, 0)$ solely in the spatial domain into cluster 2 (ball), and the remaining points of the background into cluster 1.

For each cluster, $y$ is generated as a linear function of $x$. For cluster 1, the true function is:

$$y = -x_1.$$

And for cluster 2, the true function is:

$$y = x_1.$$

After the data was generated, GPSC was applied with the following input: 2 clusters, 50 iterations with early stopping, GP input $\{x_i, s_i\}$, constant bounds $(1e^{-15}, 1e^6)$, length scale bounds $(1e^6, 1e^{15})$, input data. For the GP, the RBF kernel was used. Then K-means clustering, spectral clustering, hierarchical clustering with Ward linkage, and DBSCAN were also applied. For spectral clustering, the affinity matrix was generated with using nearest-neighbors set to 11. For DBSCAN, the maximum distance was set to 77, and minimum number of samples set to 3. The full set of $\{x_i, s_i, y_i\}$ as a vector was input into each algorithm along with

$L = 2$ clusters where relevant. For supervised C-means clustering, the supervised labels were set according to the $y$ domain with otherwise default parameters. For GDBSCAN, the covariate distance threshold was set to 3, spatial distance set to 13, and minimum set to 0. For the spatialized hierarchical clustering method, the connectivity matrix was specified using k-nearest neighbors using 1 neighbor and ward linkage. Finally, default parameters were used for Gaussian mixture model. These parameters were found by searching over a wide range of values such that the adjusted mutual information was maximized against the true labels. Parameter tuning for all methods, including GPSC, was only done on the first seed (14), and all replicates used the same set of parameters (for seeds 15-63). Any parameters not mentioned were left as default as per the scikit-learn package. The code is provided for full details and implementation.

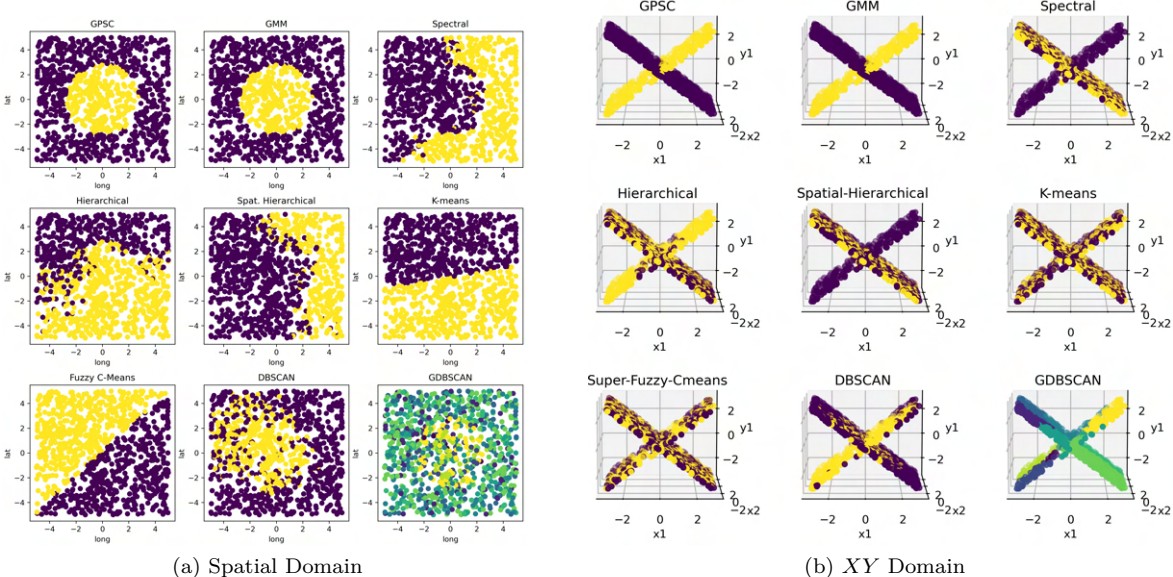

(a) Spatial Domain  (b) $XY$ Domain

Figure 9: GPSC and comparisons to spatial clustering and supervised clustering algorithms for Simulation 1.

Table 1: Adjusted Rand index and adjusted mutual information of different methods against the true labels for Simulation 1, replicated over 50 random seeds reported as mean $\pm$ standard deviation for.

| METHOD | ARI | AMI | METHOD | ARI | AMI |
|---|---|---|---|---|---|
| GPSC | $\mathbf{0.91 \pm 0.27}$ | $\mathbf{0.90 \pm 0.27}$ | GMM | $0.82 \pm 0.39$ | $0.82 \pm 0.39$ |
| K-MEANS | $0.00 \pm 0.00$ | $0.00 \pm 0.00$ | C-MEANS | $0.00 \pm 0.00$ | $0.00 \pm 0.00$ |
| HIER. | $0.03 \pm 0.03$ | $0.11 \pm 0.06$ | SPAT. HIER. | $0.03 \pm 0.03$ | $0.12 \pm 0.06$ |
| DBSCAN | $0.10 \pm 0.08$ | $0.08 \pm 0.06$ | GDBSCAN | $0.09 \pm 0.03$ | $0.24 \pm 0.04$ |
| SPECTRAL | $0.03 \pm 0.14$ | $0.15 \pm 0.12$ | | | |

## D.2 Simulation 2

The data used in this simulation takes the form $\{(s_i, x_i, y_i)\}_{i=1}^n$, where $s_i \in \mathbb{R}^2$, the spatial domain, $x_i \in \mathbb{R}^2$, the covariate domain, and $y_i \in \mathbb{R}$, the response domain, for visualization purposes.

In this simulation, both $s_i \in \mathbb{R}^2$ and $x_i \in \mathbb{R}^2$ are generated from independent uniform distributions, where $s_i \sim \text{Unif}(-5, 5)$ and $x_i \sim \text{Unif}(-3, 3)$ component-wise.

After generating the data $\{(s_i, x_i)\}_{i=1}^n$, where $n = 1000$ samples, the domain square is subdivided into again two clusters, the ring and background. Cluster 1 (ring) was made by subsetting all points $\{(s_i, x_i)\}$ within 3.5 but greater than 2 units of the point (0,0) solely in the spatial domain, with the rest forming cluster 2.

For each cluster, $y$ is generated as a nonlinear function of just $x_i$. For cluster 1, the true nonlinear function is:

$$y = -(x_1)^3.$$

And for cluster 2, the true nonlinear function is:

$$y = (x_1)^3.$$

After the data was generated, the GPSC was applied with the following input: 2 clusters, 20 iterations with early stopping, (note that GP input remains $\{x_i, s_i\}$ even though the true functions generating the clusters are functions only of $x$), constant bounds $(1e^{-15}, 1e^6)$, length scale bounds $(1e^6, 1e^{15})$, input data. For the GP, again the RBF kernel was used.

Then K-means clustering, spectral clustering, hierarchical clustering with Ward linkage, and DBSCAN was also applied. For spectral clustering, the affinity matrix was generated with using nearest-neighbors set to 5. For DBSCAN, the maximum distance was set to 3, and minimum number of samples set to 3. The full set of $\{x_i, s_i, y_i\}$ as a vector was input into each algorithm along with $L = 2$ clusters where relevant. Any parameters not mentioned were left as default as per the scikit-learn package. For spectral clustering, all neighbors between 1 and 50 were tested by comparing the adjusted mutual information scores. For DBSCAN, the maximum distance distance was tested between 1 and 100, and for each distance, the minimum samples were tested between 1 and 300, again by adjusted mutual information scores. For supervised C-means clustering, the supervised labels were set according to the $y$ domain with otherwise default parameters. For GDBSCAN, the covariate distance threshold was set to 3, spatial distance threshold set to 13, and minimum set to 0. Finally, for the spatialized hierarchical clustering method, the connectivity matrix was specified using k-nearest neighbors using 11 neighbors and ward linkage. For Gaussian mixture model, the default parameters were used. These parameters were found by searching over a wide range of values such that the adjusted mutual information was maximized against the true labels. Parameter tuning for all methods, including GPSC, was only done on the first seed (14), and all replicates used the same set of parameters (for seeds 15-63). Any parameters not mentioned were left as default as per the scikit-learn package. The code is provided for full details and implementation.

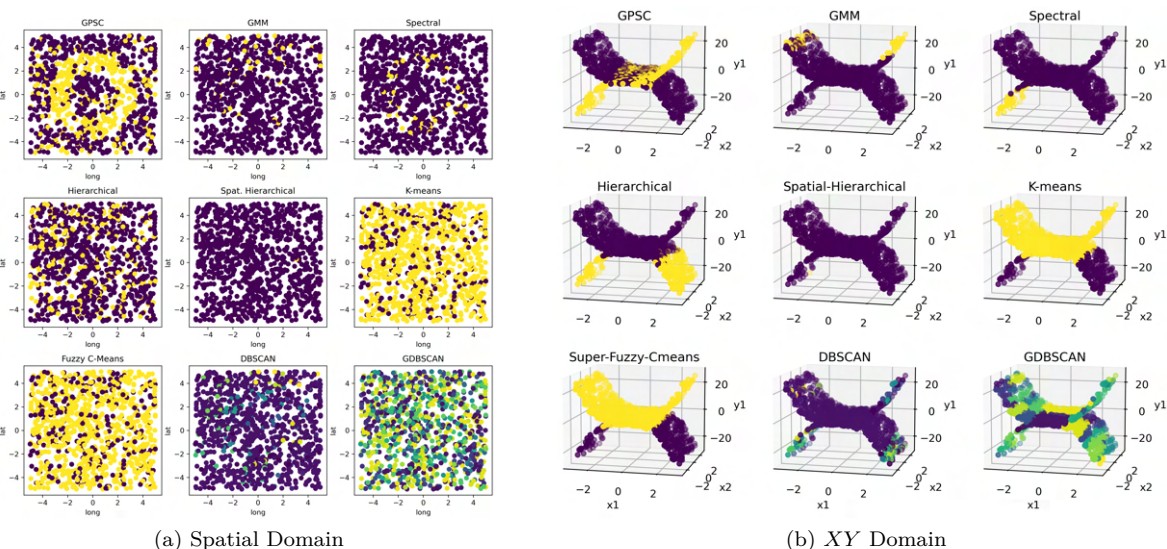

(a) Spatial Domain          (b) $XY$ Domain

Figure 10: GPSC and comparisons to spatial clustering and supervised clustering algorithms for Simulation 2.

Table 2: Adjusted Rand index and adjusted mutual information of different methods against the true labels for Simulation 2, replicated over 50 random seeds reported as mean $\pm$ standard deviation.

| METHOD | ARI | AMI | METHOD | ARI | AMI |
|---|---|---|---|---|---|
| GPSC | $\mathbf{0.38 \pm 0.13}$ | $\mathbf{0.28 \pm 0.10}$ | GMM | $0.11 \pm 0.12$ | $0.07 \pm 0.11$ |
| K-MEANS | $0.00 \pm 0.01$ | $0.00 \pm 0.00$ | C-MEANS | $0.00 \pm 0.01$ | $0.00 \pm 0.00$ |
| HIER. | $0.00 \pm 0.00$ | $0.00 \pm 0.00$ | SPAT. HIER. | $0.00 \pm 0.00$ | $0.00 \pm 0.00$ |
| DBSCAN | $0.08 \pm 0.05$ | $0.10 \pm 0.02$ | GDBSCAN | $0.02 \pm 0.01$ | $0.17 \pm 0.01$ |
| SPECTRAL | $0.02 \pm 0.09$ | $0.06 \pm 0.06$ | | | |

### D.3 Simulation 3 - Noisy Cluster Results

The data used in this simulation takes the form $\{(s_i, x_i, y_i)\}_{i=1}^n$, where $s_i \in \mathbb{R}^2$, the spatial domain, $x_i \in \mathbb{R}^2$, the covariate domain, and $y_i \in \mathbb{R}$, the response domain, for visualization purposes.

In this simulation, both $s_i \in \mathbb{R}^2$ and $x_i \in \mathbb{R}^2$ are generated from independent uniform distributions, where $s_i \sim \text{Unif}(-5, 5)$, $x_1 \sim \text{Unif}(-6, 6)$, $x_2 \sim \text{Unif}(-2, 4)$ component-wise.

After generating the data $\{(s_i, x_i)\}_{i=1}^n$, where $n = 1000$ samples, the domain square is subdivided into three clusters, the sun shape cluster, moon shape cluster, and the rest region. Cluster 1 was made by subsetting all points $\{(s_i, x_i)\}$ within 2.5 units of the point (-2.2, 2.2) solely in the spatial domain. Cluster 2 was made subsetting all points $\{(s_i, x_i)\}$ within 3 units of (1.8, -1.8) and further than 2 units apart from (1, -1), again solely in the spatial domain, with the remaining points forming cluster 3.

For each cluster, $y$ is generated as a function of just $x_i$ with independent Gaussian distributed noise $\epsilon \sim N(0, \sigma^2)$. For cluster 1, the true nonlinear function is:

$$y = 40x_1^2 - 400 + \epsilon.$$

For cluster 2, the true nonlinear function is:

$$y = -(x_1 - 8)^3 + \epsilon.$$

And for cluster 3, the true nonlinear function is:

$$y = (x_1 + 8)^3 - 20 + \epsilon.$$

After the data was generated, the GPSC was applied with the following input: 3 clusters, 40 iterations with early stopping, (note that GP input remains $\{x_i, s_i\}$ even though the true functions generating the clusters are functions only of $x$), constant bounds $(1e^{-15}, 1e^4)$, length scale bounds $(1e^6, 1e^{15})$, input data. For the GP, again the RBF kernel was used. Note that here, two forms of GPSC were used. First, standard GPSC was performed with results shown in the table. Then, GPSC with $\lambda = 75$ was used, and it was shown that the GPSC model was better able to find the clusters with this spatial penalty.

Then K-means clustering, spectral clustering, hierarchical clustering with Ward linkage, and DBSCAN was also applied. For spectral clustering, the affinity matrix was generated with using nearest-neighbors set to 12. For DBSCAN, the maximum distance was set to 41, and minimum number of samples set to 27. The full set of $\{x_i, s_i, y_i\}$ as a vector was input into each algorithm along with $k = 3$ clusters where relevant. Any parameters not mentioned were left as default as per the scikit-learn package. For supervised C-means clustering, the supervised labels were set according to the $y$ domain with otherwise default parameters. For GDBSCAN, the covariate distance threshold was set to 675, spatial distance threshold set to 5, and minimum set to 0. For the spatialized hierarchical clustering method, the connectivity matrix was specified using k-nearest neighbors using 9 neighbors and ward linkage. Finally for Gaussian mixture model, the default parameters were used. These parameters were found by searching over a wide range of values such that the adjusted mutual information was maximized against the true labels. Parameter tuning for all methods,

including GPSC, was only done on the first seed (14), and all replicates used the same set of parameters (for seeds 15-63). Any parameters not mentioned were left as default as per the scikit-learn package. The code is provided for full details and implementation.

### D.3.1 $\sigma^2 = 2$

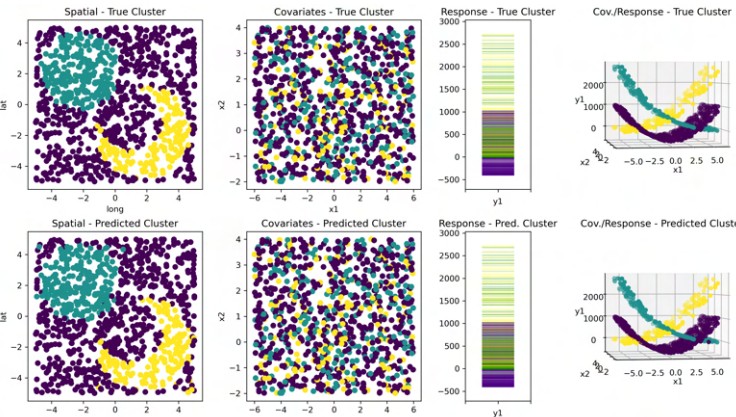

Figure 11: GPSC results for Simulation 3, $\sigma^2 = 2, L = 3$, colored by cluster and separated by data domain as in previous simulation. The first row indicates ground truth with results from GPSC in the second.

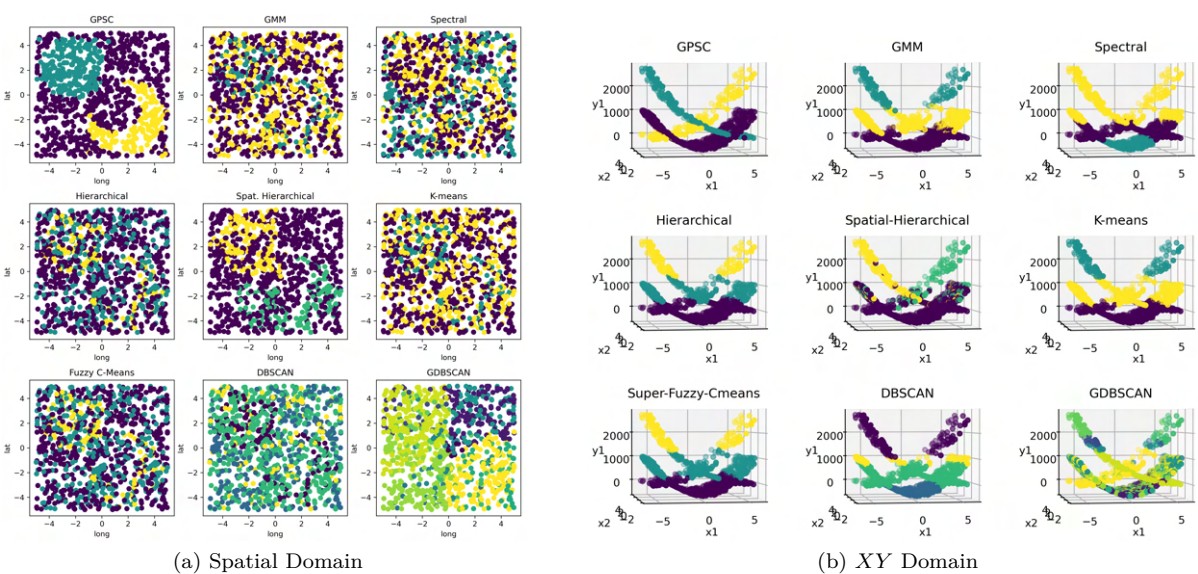

(a) Spatial Domain   (b) $XY$ Domain

Figure 12: GPSC and comparisons to spatial clustering and supervised clustering algorithms for Simulation 3, $\sigma^2 = 2, L = 3$.

Table 3: Adjusted Rand index and adjusted mutual information of different methods against the true labels for for Simulation 3, $\sigma^2 = 2, L = 3$, replicated over 50 random seeds reported as mean $\pm$ standard deviation.

| METHOD | ARI | AMI | METHOD | ARI | AMI |
|---|---|---|---|---|---|
| GPSC | $\mathbf{0.72 \pm 0.27}$ | $\mathbf{0.70 \pm 0.24}$ | GMM | $0.16 \pm 0.02$ | $0.14 \pm 0.02$ |
| K-MEANS | $0.17 \pm 0.02$ | $0.13 \pm 0.01$ | C-MEANS | $0.16 \pm 0.02$ | $0.13 \pm 0.01$ |
| HIER. | $0.17 \pm 0.03$ | $0.13 \pm 0.03$ | SPAT. HIER. | $0.16 \pm 0.11$ | $0.17 \pm 0.08$ |
| DBSCAN | $0.22 \pm 0.04$ | $0.15 \pm 0.03$ | GDBSCAN | $0.10 \pm 0.02$ | $0.24 \pm 0.04$ |
| SPECTRAL | $0.08 \pm 0.02$ | $0.16 \pm 0.02$ | | | |

### D.3.2 $\quad \sigma^2 = 50$

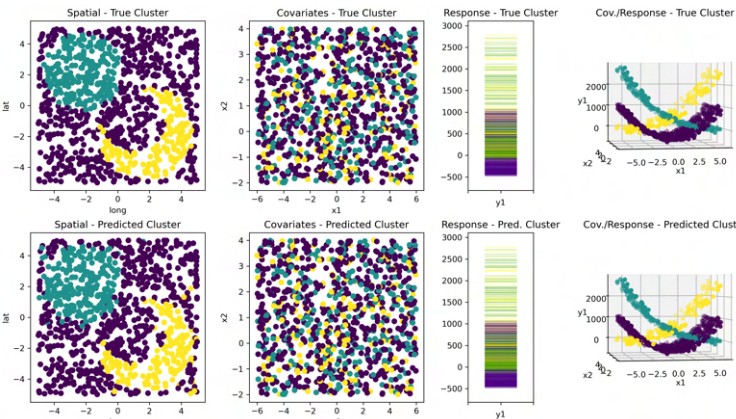

Figure 13: GPSC results for for Simulation 3, $\sigma^2 = 50, L = 3$, colored by cluster and separated by data domain as in previous simulation. The first row indicates ground truth with results from GPSC in the second.

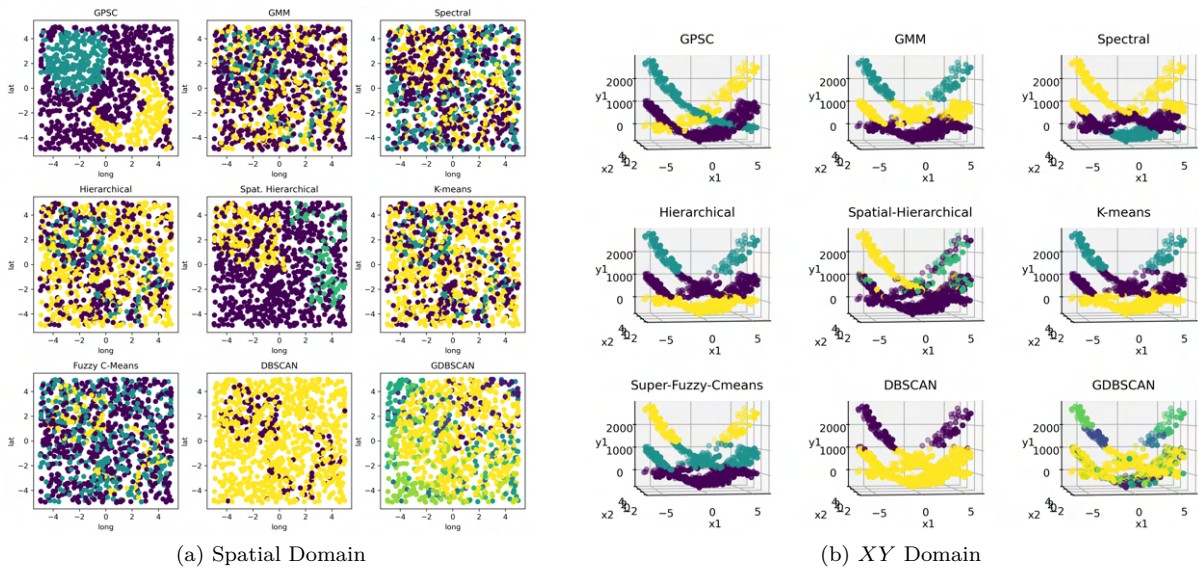

(a) Spatial Domain        (b) $XY$ Domain

Figure 14: GPSC and comparisons to spatial clustering and supervised clustering algorithms for Simulation 3, $\sigma^2 = 50, L = 3$.

Table 4: Adjusted Rand index and adjusted mutual information of different methods against the true labels for for Simulation 3, $\sigma^2 = 50, L = 3$, replicated over 50 random seeds reported as mean $\pm$ standard deviation.

| METHOD | ARI | AMI | METHOD | ARI | AMI |
|---|---|---|---|---|---|
| GPSC | $\mathbf{0.73 \pm 0.25}$ | $\mathbf{0.71 \pm 0.22}$ | GMM | $0.16 \pm 0.02$ | $0.13 \pm 0.03$ |
| K-MEANS | $0.17 \pm 0.02$ | $0.13 \pm 0.01$ | C-MEANS | $0.16 \pm 0.02$ | $0.13 \pm 0.01$ |
| HIER. | $0.17 \pm 0.03$ | $0.13 \pm 0.03$ | SPAT. HIER. | $0.17 \pm 0.10$ | $0.18 \pm 0.08$ |
| DBSCAN | $0.23 \pm 0.03$ | $0.13 \pm 0.03$ | GDBSCAN | $0.09 \pm 0.03$ | $0.23 \pm 0.03$ |
| SPECTRAL | $0.08 \pm 0.02$ | $0.16 \pm 0.01$ | | | |

### D.3.3 $\sigma^2 = 100$

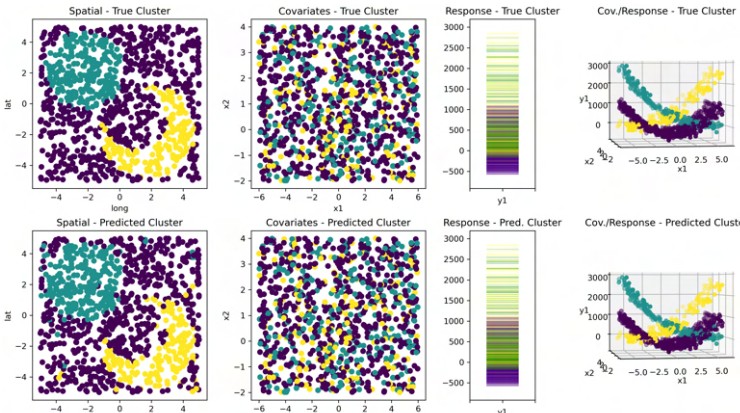

Figure 15: GPSC results for for Simulation 3, $\sigma^2 = 100, L = 3$, colored by cluster and separated by data domain as in previous simulation. The first row indicates ground truth with results from GPSC in the second.

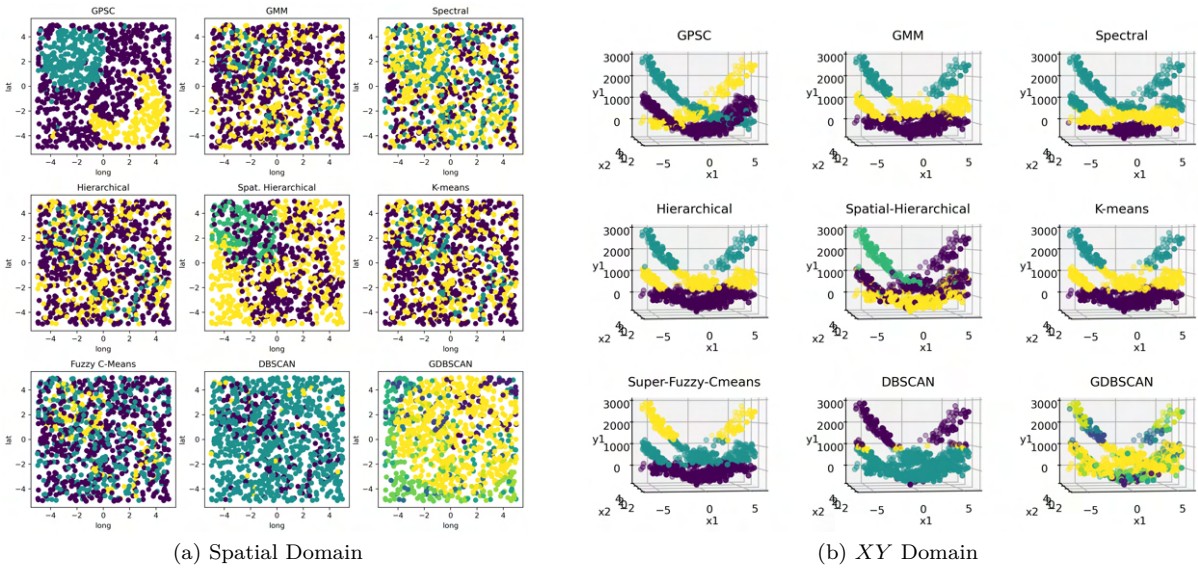

(a) Spatial Domain          (b) $XY$ Domain

Figure 16: GPSC and comparisons to spatial clustering and supervised clustering algorithms for Simulation 3, $\sigma^2 = 100, L = 3$.

Table 5: Adjusted Rand index and adjusted mutual information of different methods against the true labels for for Simulation 3, $\sigma^2 = 100, L = 3$, replicated over 50 random seeds reported as mean $\pm$ standard deviation.

| METHOD | ARI | AMI | METHOD | ARI | AMI |
|---|---|---|---|---|---|
| GPSC | $\mathbf{0.56 \pm 0.26}$ | $\mathbf{0.55 \pm 0.23}$ | GMM | $0.15 \pm 0.02$ | $0.13 \pm 0.03$ |
| K-MEANS | $0.17 \pm 0.02$ | $0.13 \pm 0.01$ | C-MEANS | $0.16 \pm 0.02$ | $0.13 \pm 0.01$ |
| HIER. | $0.16 \pm 0.04$ | $0.13 \pm 0.03$ | SPAT. HIER. | $0.15 \pm 0.09$ | $0.17 \pm 0.06$ |
| DBSCAN | $0.21 \pm 0.02$ | $0.10 \pm 0.02$ | GDBSCAN | $0.09 \pm 0.03$ | $0.23 \pm 0.04$ |
| SPECTRAL | $0.07 \pm 0.02$ | $0.14 \pm 0.01$ | | | |

### D.3.4   $\sigma^2 = 200$

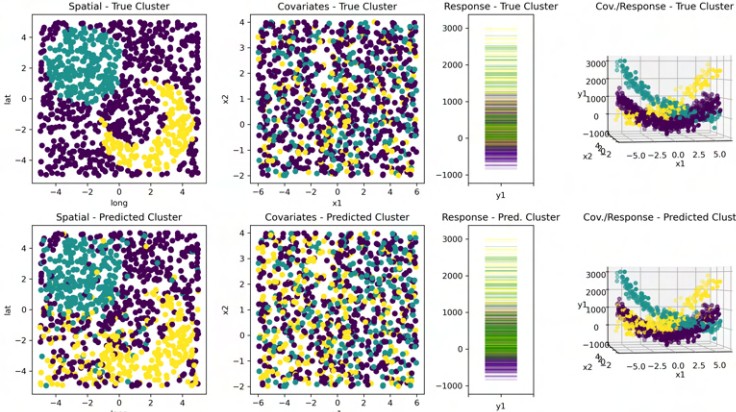

Figure 17: GPSC results for for Simulation 3, $\sigma^2 = 200, L = 3$, colored by cluster and separated by data domain as in previous simulation. The first row indicates ground truth with results from GPSC in the second.

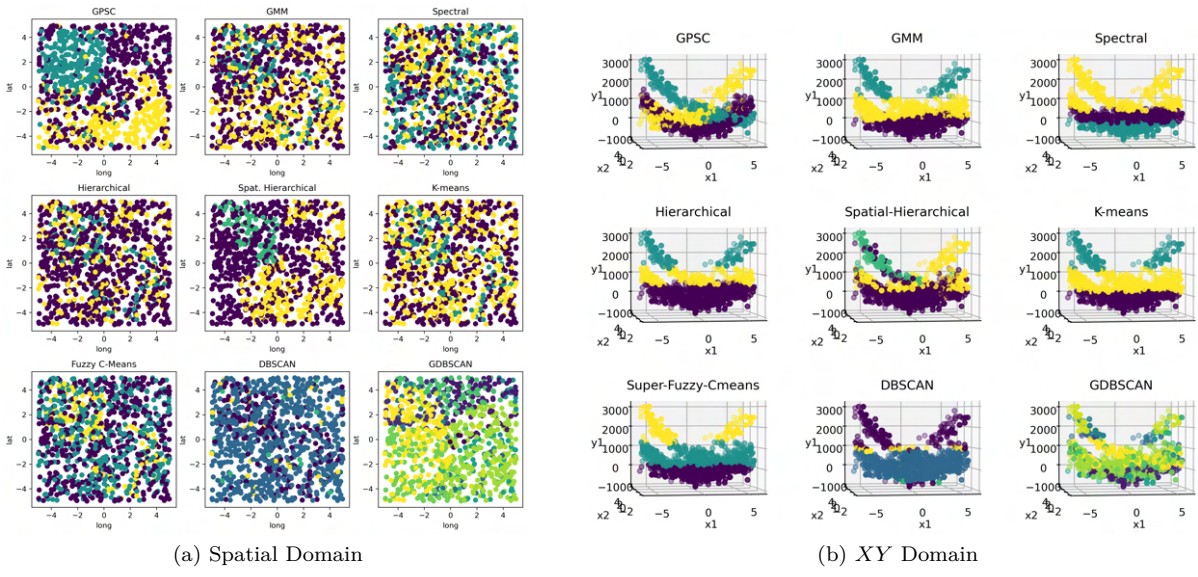

(a) Spatial Domain          (b) $XY$ Domain

Figure 18: GPSC and comparisons to spatial clustering and supervised clustering algorithms for Simulation 3, $\sigma^2 = 200, L = 3$.

Table 6: Adjusted Rand index and adjusted mutual information of different methods against the true labels for for Simulation 3, $\sigma^2 = 200, L = 3$, replicated over 50 random seeds reported as mean $\pm$ standard deviation.

| METHOD | ARI | AMI | METHOD | ARI | AMI |
|---|---|---|---|---|---|
| GPSC | $\mathbf{0.33 \pm 0.17}$ | $\mathbf{0.33 \pm 0.15}$ | GMM | $0.16 \pm 0.02$ | $0.13 \pm 0.02$ |
| K-MEANS | $0.17 \pm 0.02$ | $0.13 \pm 0.01$ | C-MEANS | $0.16 \pm 0.02$ | $0.13 \pm 0.01$ |
| HIER. | $0.16 \pm 0.04$ | $0.12 \pm 0.02$ | SPAT. HIER. | $0.15 \pm 0.08$ | $0.1 \pm 0.06$ |
| DBSCAN | $0.16 \pm 0.02$ | $0.06 \pm 0.01$ | GDBSCAN | $0.08 \pm 0.02$ | $0.22 \pm 0.03$ |
| SPECTRAL | $0.06 \pm 0.02$ | $0.10 \pm 0.01$ | | | |

## D.4 Simulation 3 - Cluster Overspecification Results

### D.4.1 $L = 3, \sigma^2 = 2$

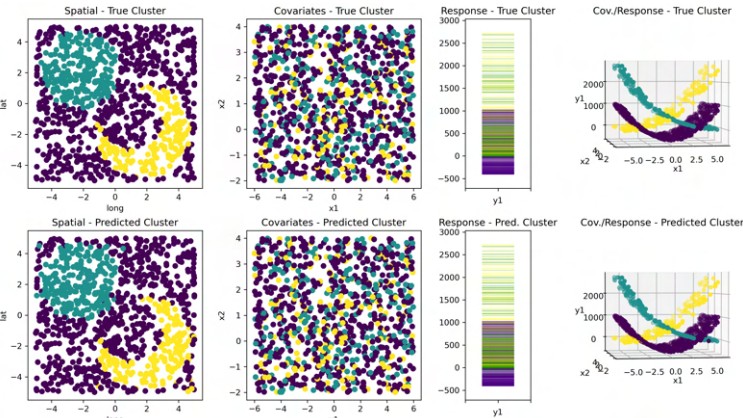

Figure 19: GPSC results for for Simulation 3, $L = 3, \sigma^2 = 2$, colored by cluster and separated by data domain as in previous simulation. The first row indicates ground truth with results from GPSC in the second.

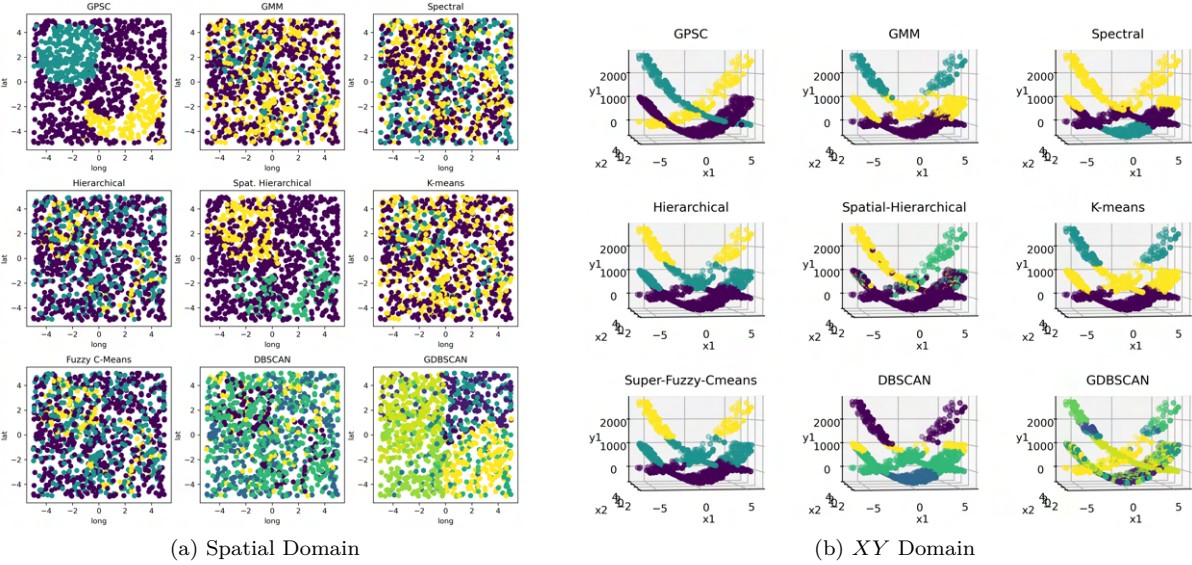

(a) Spatial Domain                    (b) $XY$ Domain

Figure 20: Comparisons to spatial and supervised clustering algorithms for Simulation 3, $L = 3, \sigma^2 = 2$.

Table 7: Adjusted Rand index and adjusted mutual information of different methods against the true labels for for Simulation 3, $L = 3, \sigma^2 = 2$, replicated over 50 random seeds reported as mean $\pm$ standard deviation.

| METHOD | ARI | AMI | METHOD | ARI | AMI |
|---|---|---|---|---|---|
| GPSC | $\mathbf{0.72 \pm 0.27}$ | $\mathbf{0.70 \pm 0.24}$ | GMM | $0.16 \pm 0.02$ | $0.14 \pm 0.02$ |
| K-MEANS | $0.17 \pm 0.02$ | $0.13 \pm 0.01$ | C-MEANS | $0.16 \pm 0.02$ | $0.13 \pm 0.01$ |
| HIER. | $0.17 \pm 0.03$ | $0.13 \pm 0.03$ | SPAT. HIER. | $0.16 \pm 0.11$ | $0.17 \pm 0.08$ |
| DBSCAN | $0.22 \pm 0.04$ | $0.15 \pm 0.03$ | GDBSCAN | $0.10 \pm 0.02$ | $0.24 \pm 0.04$ |
| SPECTRAL | $0.08 \pm 0.02$ | $0.16 \pm 0.02$ | | | |

**D.4.2**  $L = 4, \sigma^2 = 2$

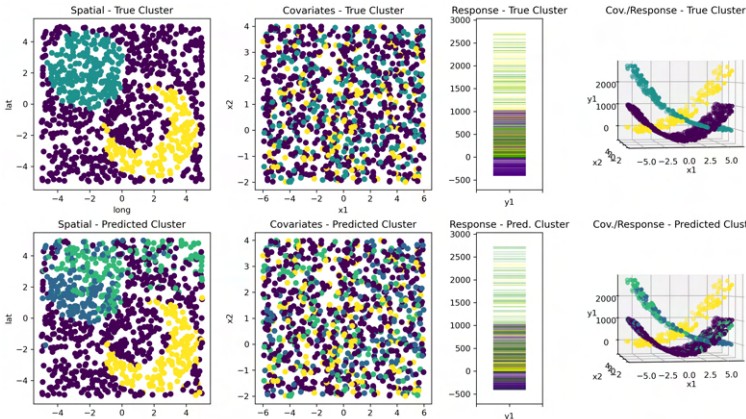

Figure 21: GPSC results for for Simulation 3, $L = 4, \sigma^2 = 2$, colored by cluster and separated by data domain as in previous simulation. The first row indicates ground truth with results from GPSC in the second.

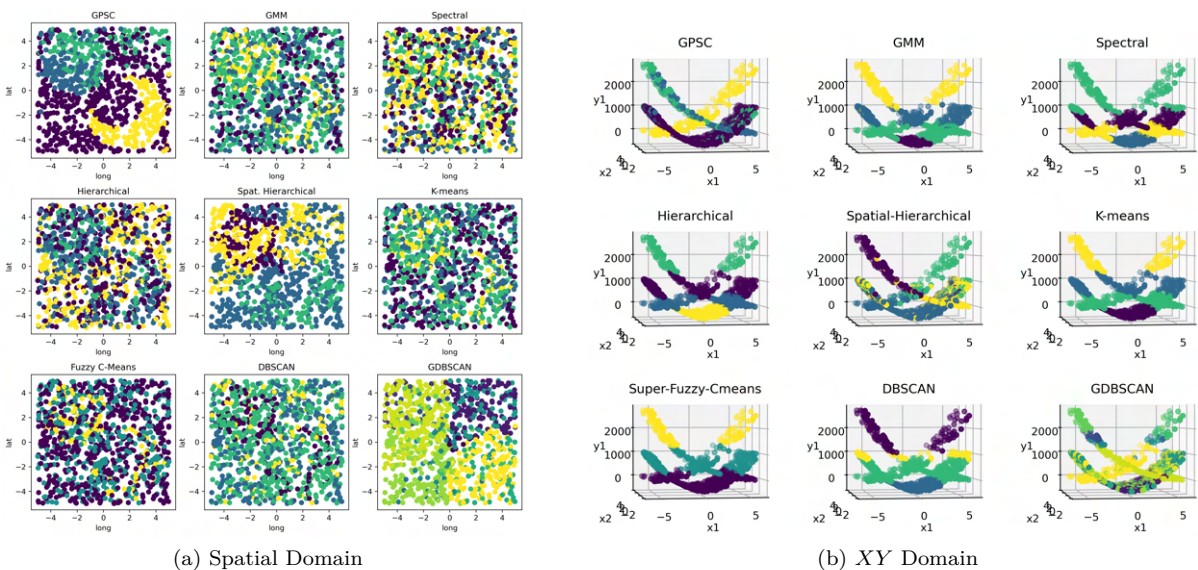

(a) Spatial Domain  (b) $XY$ Domain

Figure 22: GPSC and comparisons to spatial clustering and supervised clustering algorithms, $L = 4, \sigma^2 = 2$.

Table 8: Adjusted Rand index and adjusted mutual information of different methods against the true labels for for Simulation 3, $L = 4, \sigma^2 = 2$, replicated over 50 random seeds reported as mean $\pm$ standard deviation.

| METHOD | ARI | AMI | METHOD | ARI | AMI |
|---|---|---|---|---|---|
| GPSC | $\mathbf{0.58 \pm 0.14}$ | $\mathbf{0.69 \pm 0.11}$ | GMM | $0.08 \pm 0.01$ | $0.18 \pm 0.02$ |
| K-MEANS | $0.15 \pm 0.02$ | $0.22 \pm 0.02$ | C-MEANS | $0.16 \pm 0.02$ | $0.13 \pm 0.01$ |
| HIER. | $0.14 \pm 0.03$ | $0.19 \pm 0.04$ | SPAT. HIER. | $0.13 \pm 0.10$ | $0.20 \pm 0.07$ |
| DBSCAN | $0.22 \pm 0.04$ | $0.15 \pm 0.03$ | GDBSCAN | $0.10 \pm 0.02$ | $0.24 \pm 0.04$ |
| SPECTRAL | $0.09 \pm 0.01$ | $0.14 \pm 0.15$ | | | |

### D.4.3 $L = 5, \sigma^2 = 2$

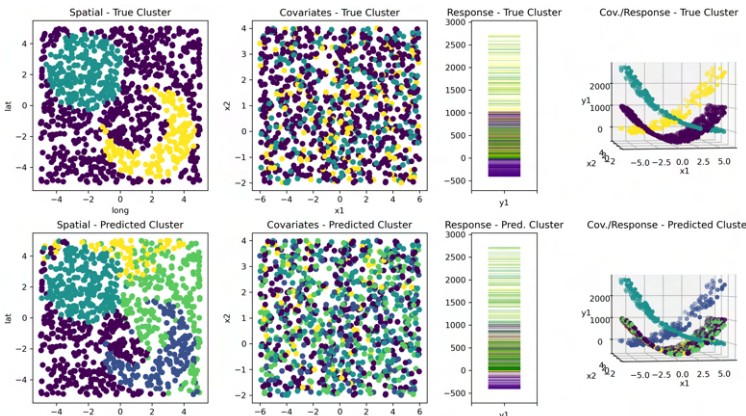

Figure 23: GPSC results for for Simulation 3, $L = 5, \sigma^2 = 2$, colored by cluster and separated by data domain as in previous simulation. The first row indicates ground truth with results from GPSC in the second.

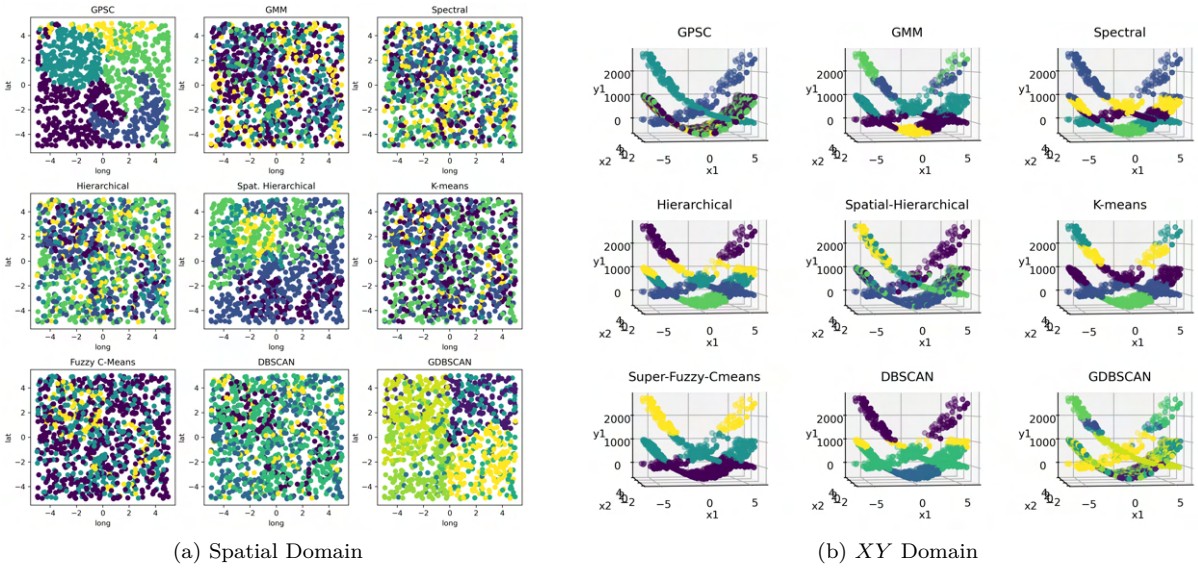

(a) Spatial Domain                    (b) $XY$ Domain

Figure 24: GPSC and comparisons to spatial clustering and supervised clustering algorithms, $L = 5, \sigma^2 = 2$.

Table 9: Adjusted Rand index and adjusted mutual information of different methods against the true labels for for Simulation 3, $L = 5, \sigma^2 = 2$, replicated over 50 random seeds reported as mean $\pm$ standard deviation.

| METHOD | ARI | AMI | METHOD | ARI | AMI |
|---|---|---|---|---|---|
| GPSC | $\mathbf{0.47 \pm 0.08}$ | $\mathbf{0.65 \pm 0.09}$ | GMM | $0.10 \pm 0.04$ | $0.20 \pm 0.06$ |
| K-MEANS | $0.15 \pm 0.02$ | $0.20 \pm 0.02$ | C-MEANS | $0.16 \pm 0.02$ | $0.13 \pm 0.01$ |
| HIER. | $0.14 \pm 0.03$ | $0.20 \pm 0.03$ | SPAT. HIER. | $0.13 \pm 0.09$ | $0.22 \pm 0.06$ |
| DBSCAN | $0.22 \pm 0.04$ | $0.15 \pm 0.03$ | GDBSCAN | $0.10 \pm 0.02$ | $0.24 \pm 0.04$ |
| SPECTRAL | $0.09 \pm 0.01$ | $0.14 \pm 0.02$ | | | |

**D.4.4** $L = 6, \sigma^2 = 2$

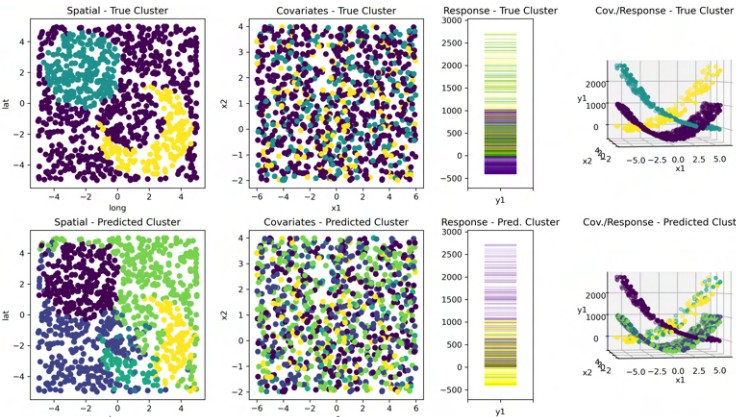

Figure 25: GPSC results for for Simulation 3, $L = 6, \sigma^2 = 2$, colored by cluster and separated by data domain as in previous simulation. The first row indicates ground truth with results from GPSC in the second.

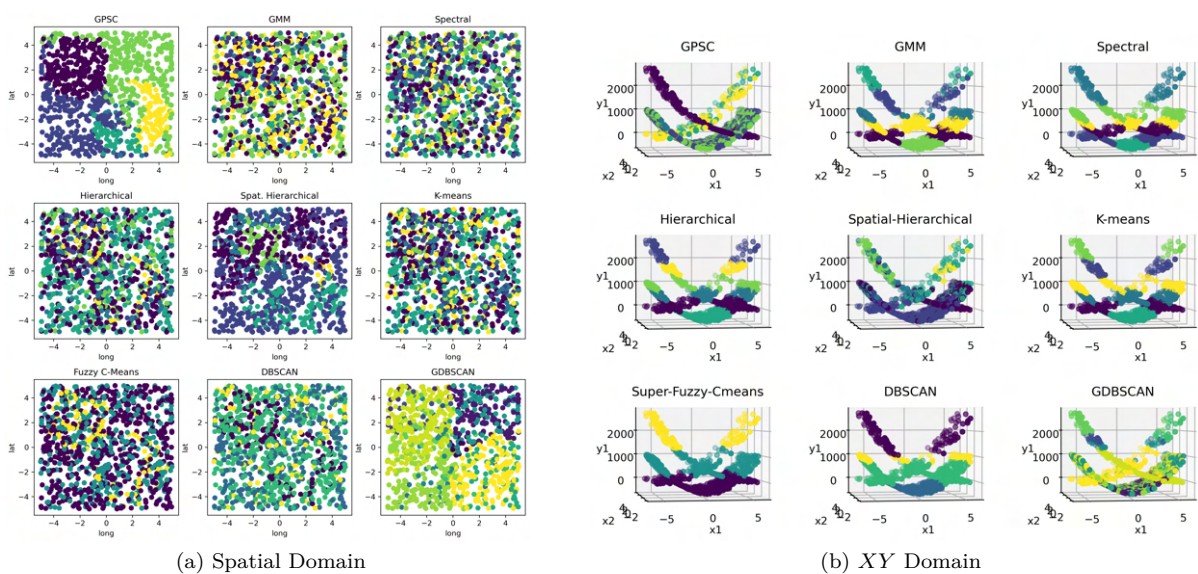

(a) Spatial Domain        (b) $XY$ Domain

Figure 26: GPSC and comparisons to spatial clustering and supervised clustering algorithms for Simulation 3, $L = 6, \sigma^2 = 2$.

Table 10: Adjusted Rand index and adjusted mutual information of different methods against the true labels for for Simulation 3, $L = 6, \sigma^2 = 2$, replicated over 50 random seeds reported as mean $\pm$ standard deviation.

| METHOD | ARI | AMI | METHOD | ARI | AMI |
|--------|-----|-----|--------|-----|-----|
| GPSC | $\mathbf{0.43 \pm 0.05}$ | $\mathbf{0.63 \pm 0.04}$ | GMM | $0.10 \pm 0.03$ | $0.24 \pm 0.04$ |
| K-MEANS | $0.14 \pm 0.02$ | $0.19 \pm 0.01$ | C-MEANS | $0.16 \pm 0.16$ | $0.13 \pm 0.01$ |
| HIER. | $0.14 \pm 0.04$ | $0.19 \pm 0.02$ | SPAT. HIER. | $0.13 \pm 0.08$ | $0.23 \pm 0.06$ |
| DBSCAN | $0.22 \pm 0.04$ | $0.15 \pm 0.03$ | GDBSCAN | $0.10 \pm 0.02$ | $0.24 \pm 0.04$ |
| SPECTRAL | $0.08 \pm 0.01$ | $0.18 \pm 0.02$ | | | |

## D.5 Simulation 3 - Functions of $s$ and $x$

The set up here is exactly as in Simulation 3, however, the functions are now functions of both the spatial domain and the covariate domain. It can be see that GPSC is still able to recover the true clusters under these conditions. Exact implementation and final parameters can be found in the submitted code.

For each cluster, $y$ is generated as a function of just $x_i$ with independent Gaussian distributed noise $\epsilon \sim N(0, 2)$. For cluster 1, the true nonlinear function is:

$$y = 10(s_1 + s_2)^2 + 40(x_1)^2 + (x_2)^2 - 500 + \epsilon.$$

For cluster 2, the true nonlinear function is:

$$y = 10(s_2)^2 - (x_1 - 8)^3 + (x_2)^3 + \epsilon.$$

For cluster 3, the true nonlinear function is:

$$y = -10(s_1)^2 + (x_1 + 8)^3 + (x_2)^3 - 20 + \epsilon.$$

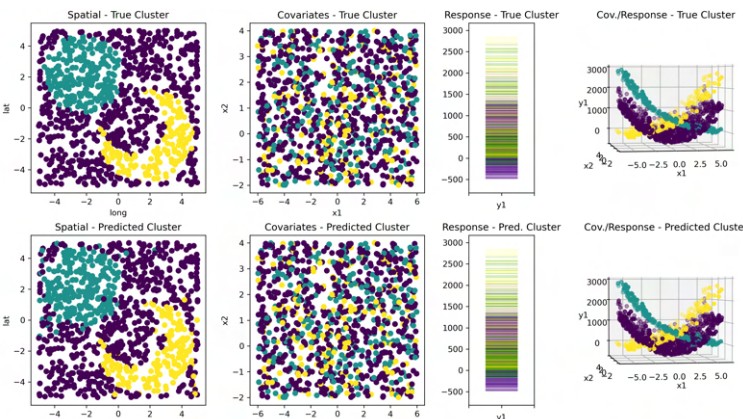

Figure 27: GPSC results for Simulation 3 analog with functions of both $s$ and $x$, colored by cluster and separated by data domain as in previous simulation. The first row indicates ground truth with results from GPSC in the second.

As can be seen, GPSC performs well regardless of whether the functional relationships are based on $s$, $x$ or $s$ and $x$. Regardless of the which case the true functional relationship is in, the full vector $(s, x, y)$ is used as input and GPSC is able to accurately recover the clusters.

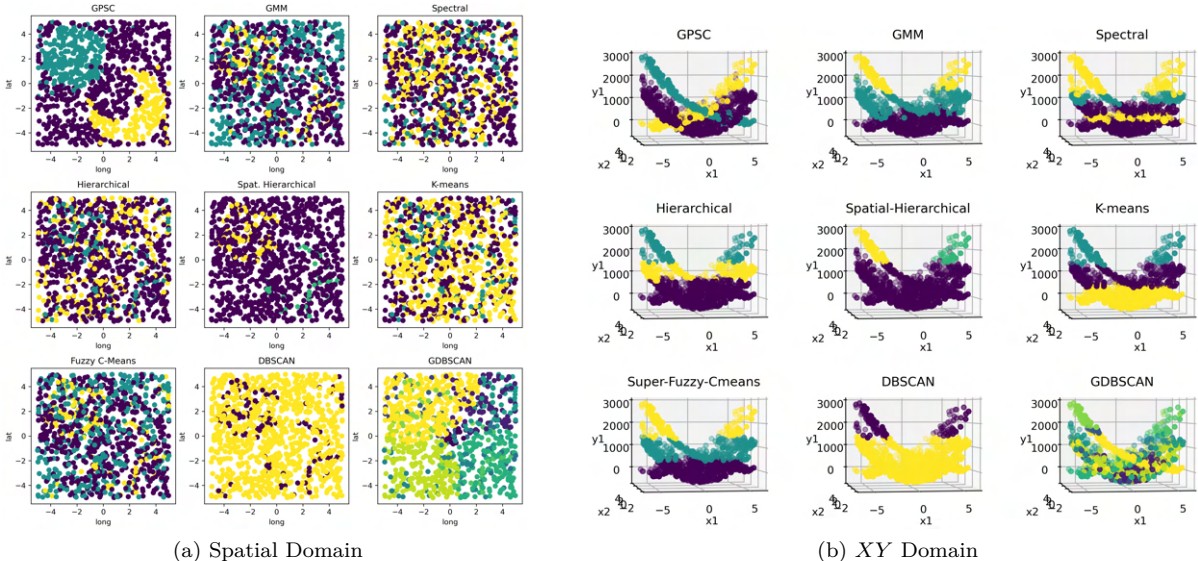

(a) Spatial Domain

(b) $XY$ Domain

Figure 28: GPSC and comparisons to spatial clustering and supervised clustering algorithms for Simulation 3 analog with functions of both $s$ and $x$.

Table 11: Adjusted Rand index and adjusted mutual information of different methods against the true labels for Simulation 3 analog with functions of both $s$ and $x$, replicated over 50 random seeds reported as mean $\pm$ standard deviation.

| METHOD | ARI | AMI | METHOD | ARI | AMI |
|---|---|---|---|---|---|
| GPSC | $\mathbf{0.69 \pm 0.28}$ | $\mathbf{0.65 \pm 0.25}$ | GMM | $0.09 \pm 0.02$ | $0.09 \pm 0.02$ |
| K-MEANS | $0.08 \pm 0.01$ | $0.05 \pm 0.01$ | C-MEANS | $0.08 \pm 0.01$ | $0.05 \pm 0.01$ |
| HIER. | $0.08 \pm 0.02$ | $0.05 \pm 0.02$ | SPAT. HIER. | $0.10 \pm 0.06$ | $0.10 \pm 0.07$ |
| DBSCAN | $0.15 \pm 0.02$ | $0.09 \pm 0.02$ | GDBSCAN | $0.09 \pm 0.02$ | $0.21 \pm 0.03$ |
| SPECTRAL | $0.01 \pm 0.05$ | $0.03 \pm 0.02$ | | | |

# E   Details on NC Tracts Data

### E.1   Application Details

The full list of variables used in the clustering analysis is as follows:

| Variable | Description |
|---|---|
| **Spatial Data S** | |
| LATITUDE | Latitude coordinate of |
| | population-weighted geographic center of tract |
| LONGITUDE | Longitude coordinate of |
| | population-weighted geographic center of tract |
| **Covariates X** | |
| PRFL_M | Men in professional occupation |
| PRFL_F | Women in professional occupation |
| LS_HS | Less than high school education |
| SINGLE | Single with dependent |
| HSHLDR_F | Female head of household |
| NHBLK | Non-Hispanic Black |
| PA | Public assistance |
| POV | Poverty |
| NO_VHCL | No vehicle |
| RENT | Rental housing |
| CROWD | Crowded housing |
| UNMPLYD | Unemployment |
| PHONE | No phone |
| ACET | Acetaldehyde |
| BENZENE | Benzene |
| BUTA | 1,3-Butadiene |
| CARBON | Carbon Tetrachloride |
| DIESEL | Diesel PM2.5 |
| ETHYL | Ethylbenzene |
| FORM | Formaldehyde |
| HEXANE | Hexane |
| LEAD | Lead compounds |
| MANG | Manganese compounds |
| MERC | Mercury compounds |
| METH | Methanol |
| METHYL | Methyl Chloride |
| NICK | Nickel |
| TOLUENE | Toluene |
| XYLENE | Xylenes |
| **Response Y** | |
| MLCJOINT | Overall class membership into 8 possible groups |

Table 12: Full set of variables used for NC tracts data application.

Again, the analysis was performed on an Apple Macbook Pro with M1 Pro processor with 32 GB of memory. The scikit-learn clustering package Pedregosa et al. (2011) package was used for all experiments to perform comparison K-means clustering as well as handling Gaussian process modeling for the GPSC algorithm and computing clustering metrics. Although we are unable to release the data and auxiliary files for our real world application, the code for clustering and plotting has been submitted along with all the simulation code. For both K-means and GPSC, the full set of data shown in Table 12 including the covariates and spatial data were input into both algorithms.

### E.2 Additional Real World Application Comparisons

In this section we present additional results of the best performing competitors from the simulation studies (using default parameters) on the CBCS real world example of the main paper, which already contained the comparison to K-means clustering. We also include one algorithm of each type from the set of competitors, again for diversity of results. It can be seen that the different types of clustering models have distinct differences to the results of GPSC as discussed below.

### E.2.1 Gaussian Mixture Model

Here we report the clustering results of the Gaussian Mixture Model. It can be seen that the results are visibly similar to the results of K-means clustering, where again the algorithm appears to mostly center the cluster diversity around the major urban centers of the state, with fewer cluster diversity across the extremities and regions between the urban centers.

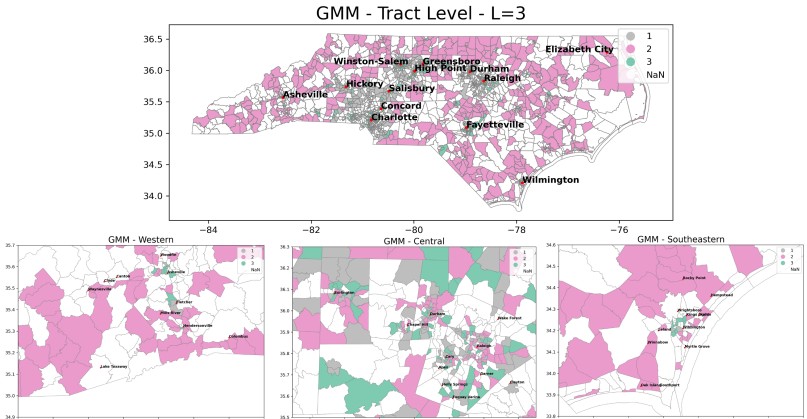

Figure 29: GMM results for the real world application presented in main paper.

### E.2.2 Spectral Clustering

Spectral clustering, similar to the spatial hierarchical clustering results presented below, seems to pick up more global trends with lower nuance specifically around the dense city regions of the state.

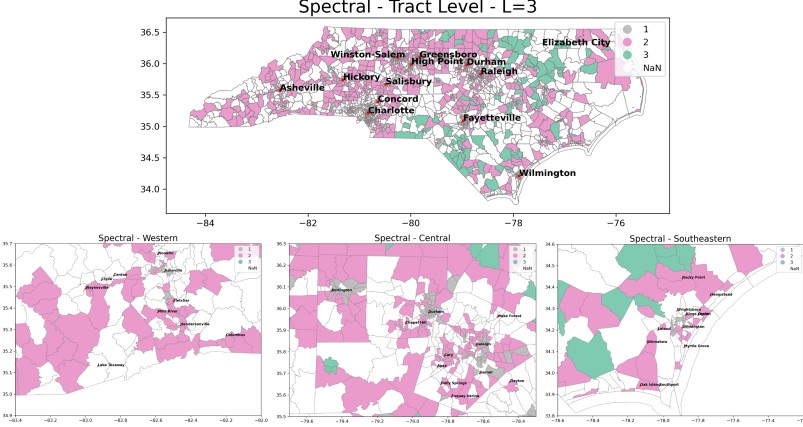

Figure 30: Spectral clustering results for the real world application presented in main paper.

### E.2.3   Spatial Hierarchical Clustering

Spatial hierarchical clustering is presented here with 5 neighbors (result did not vary significantly over different specifications of the neighbor count). It can be seen that although the algorithm may be picking up on more global trends across the state, there is decreased nuance around the specific city centers of the state.

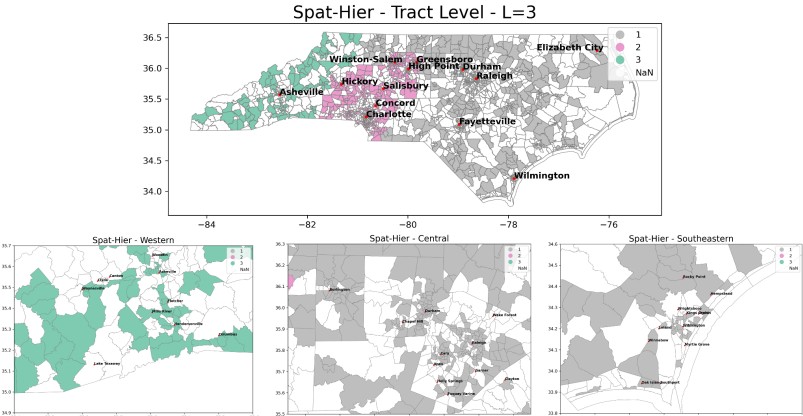

Figure 31: Spatial hierarchical clustering results for the real world application presented in main paper.

### E.2.4   DBSCAN

Here DBSCAN was chosen over GDBSCAN due to having fewer hyperparameters required to tune (default used), while having similar performance in the simulation studies. It can be seen here that the main challenge of DBSCAN (as well as GDBSCAN) is the inability to mandate the number of clusters, especially in this application where we specifically seek a small number of clusters for interpretability.

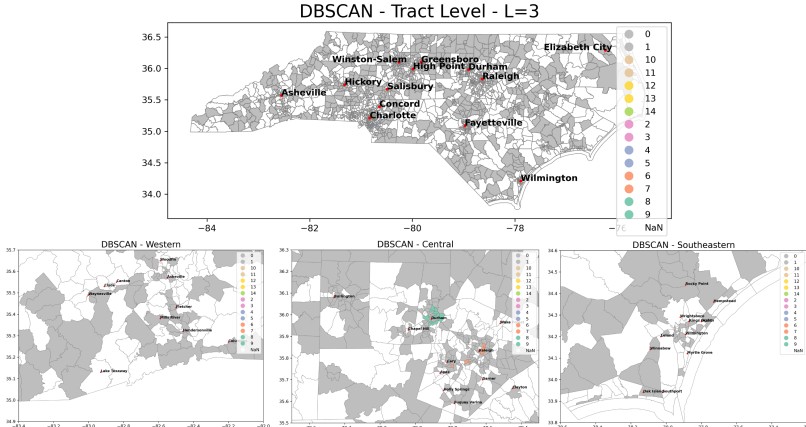

Figure 32: DBSCAN clustering results for the real world application in the main paper.

