# OpenReview forum: "Gaussian Process Spatial Clustering"
_TMLR — Withdrawn by Authors_

### Review · Reviewer_rqQZ · 2024-05-13

**Summary Of Contributions:**

The authors consider the task of clustering instances that are characterised by a set consisting of spatial location (2D coordinate), inputs features $x$ and output $y$. The task is to cluster the instances so that the relationship between $x$ (or combination of $s$ and $x$) and $y$ is similar within a cluster. The task is motivated by one specific example that is also used for demonstrating the method. The methodological content corresponds to definition of the model, its theoretical analysis, and some illustrations on simulated data.

**Audience:**

No

**Claims And Evidence:**

No

**Requested Changes:**

I do no see how this could be adapted for a publication in TMLR, given that the work is completely detached from the literature in the field and the proposed method is obvious and largely a special case of a family of methods that have been studied for 25 years.

**Strengths And Weaknesses:**

The motivational need for improves spatial clustering tools is nice and it is good the work is grounded on a real example scenario, but the paper considers one very specific instance within the broader task formulation and I do not see particularly clear connection between the high-level motivation and the specific instance. Furthermore, as will be detailed later, the method itself if is straightforward and effectively a special case of a broadly studied family of methods, yet the authors do not discuss the previous work in any way.

The artificial data examples resemble the illustrations in the early works from roughly two decades ago and are not likely to be interesting for readers within machine learning field, and in general the paper is completely detached from the modern literature. The only references to scientific articles published during the last decade or so are for reviews and articles in cancer research, meaning there are *no* citations to any new machine learning content published during the last decade. As such, the paper is clearly not publishable in a machine learning venue as it will be of no interest for the core audience, and hence potential publication venues should be search for within the application area.

Briefly on the main issues:

- The method is effectively a special case of 'mixture of GPs', originally published by Tresp (Mixtures of Gaussian Processes, NeurIPS 2000), and discussed in numerous papers since then. The mixture formulation is more flexible and the hard clustering considered here, but this method could be obtained as a special case. Even if not considering this a weakness, the related work on mixtures should naturally be covered.

- The core technical development is on clustering regression functions, which is not directly related to the original task of spatial clustering. If I understood right, the spatial coordinates are simply considered as extra features that removes the whole point on treating this as a spatial problem. Section 2 hints at an extension that encourages spatial continuity, but this is not used in practice.

- The motivation of why exactly we would want to use similarity of regression functions as the clustering criterion is vague or missing.

- The simulation studies are simplifications of the broad literature on various manifold clustering works that were a hot topic during the first decade of the millennium, with several clustering-based solutions proposed for these kinds of setups, for both linear and non-linear cases. The results are largely obvious and the baselines used here are not really sensible; why would we want to consider standard clustering algorithms here when they are not designed for this task but various other methods would be?

- The real application is quite simplified, considering only one instance with only three clusters and showing that there is actually not that much of a difference between the proposed method and the simples possible algorithm one could consider, the k-means clustering.

---

### Review · Reviewer_WKKT · 2024-05-24

**Summary Of Contributions:**

The authors propose the Gaussian Process (GP) Spatial Clustering, which starts from a random clustering and refines it based on GP regression models learned for each cluster separately. The refinement step is based on reassigning observations to clusters providing best mapping from inputs to the output based on the least squares error.

The approach is evaluated on synthetic datasets with ground truth and compared here to a variety of standard clustering procedures. The approach is further investigated on the census tracts of North Carolina (NC)  and found to provide meaningful representations in terms of the extracted clusters.

**Audience:**

Yes

**Broader Impact Concerns:**

This is well covered in the article.

**Claims And Evidence:**

Yes

**Requested Changes:**

Strengthened real experimentation using more real datasets and comparisons to existing methodologies including GPMoE and GPLVM based approaches.

Suggestions for the tuning of number of clusters or at least investigating how interpretations are impacted by these choices.

**Strengths And Weaknesses:**

Strengths:

The approach is straightforward to implement and can exploit any standard GPR implementation.

The results outperforms conventional clustering by exploiting the supervised structure in the output from inputs to outputs as learned by the GPR, which on synthetic and real data appears useful.

The approach has provided mathematical properties in regards to convergence to correct clustering.

Weaknesses
The approach is very incremental and the strength of the methodology lies in the theoretical derivations whereas the experimental analysis on real data is very limited only considering one dataset and here only compared to a very simple k-means based procedure. The experimentation could thus be substantially improved.

Furthermore, I think the relations to a very related approach named GPMoE, i.e. Gaussian Process Mixture of Expert modeling procedure is completely missing while highly relevant to compare against. Notably, mixture of expert models segments the input space using clusters and use these clusters to define separate expert models, i.e. cluster specific Gaussian Process Regression models, see also:
https://proceedings.neurips.cc/paper_files/paper/2001/file/9afefc52942cb83c7c1f14b2139b09ba-Paper.pdf
https://proceedings.neurips.cc/paper_files/paper/2008/hash/f4b9ec30ad9f68f89b29639786cb62ef-Abstract.html
This in the traditional sense corresponds to combining GPs with GMMs but more advanced mixture modeling procedures has here also been considered, see for instance also:
https://www.gatsby.ucl.ac.uk/~ucgtcbl/papers/BluTehHel2011.pdf
(i.e., for results see Figure 1.9)

I am very much in doubt what the benefits of the proposed approach compared to these existing GPMoE models are – clearly, the clustering can be done in a highly non-linear non-parametric way as it is based on assigning using the outputs rather than segmenting the input space according to expert specific models, but in the end the authors need to carefully relate their approach here to such alternative and well established approaches.

The approach further needs to be contrasted GPLVMs which has also been advanced to account for clustering, see for instance:
https://www.hindawi.com/journals/complexity/2021/8864981/

Also determining the number of components is here left unanswered which is a weak-point of the paper as knowing how many clusters to use is typically very unclear. The proposed approach is thus not very solid in relying on specifying a priori the number of clusters as opposed to a principled framework enabling the learning of the number of clusters (as is viable in GPMoE).

---

### Review · Reviewer_b7MA · 2024-05-26

**Summary Of Contributions:**

The authors propose a GPSC (gaussian process spatial clustering) which models spatially organized data (s, x, y) where s is a lat/long, x is a set of features (e.g. demographic or socioeconomic  features) and y is some response variable.

The proposed GPSC model posits that there exists some clustering/partitioning of the domain such that the function that maps (s, x) -> y is drawn from some Gaussian process. They propose a simple algorithm for clustering a set of points while fitting a GP to each individual cluster — this algorithm looks a lot like k-means (hard EM) where it alternates between assigning points to clusters, and fitting a GP to each cluster.

There is a theoretical analysis of the algorithm bounding the number of cluster assignment errors at a given iteration.  The authors also demonstrate the success of their algorithm empirically with many simulations as well as an application to a real world dataset where the task is to cluster “tracts” from North Carolina based on spatial coordinates, socioeconomic and environmental variables.

**Audience:**

Yes

**Claims And Evidence:**

No

**Requested Changes:**

* In its current form, the paper is not written clearly enough for me to advocate for acceptance. I would ask the authors to clarify exactly how their approach treats spatial coordinates separately from other (e.g. demographic) covariates as well as to fix any mistakes in the first paragraph of Sec 2.2.  As it is written, I’m not convinced that this is truly a spatial clustering paper.
* Adding a second or third real world dataset would significantly strengthen this work.

**Strengths And Weaknesses:**

**Strengths**:
* Algorithm 1 (the main algorithm) is conceptually simple (and looks basically like a hard EM style algorithm)
* The authors provide a theoretical analysis of the approach.
* The model should also combine easily with other complementary works that scale up GP regression so as to be run on larger datasets.
* Finally the authors present quite extensive simulations showing when their approach is effective.

**Weaknesses**:

After reading this paper it is unclear to me if spatial coordinates are actually treated in some separate way even though it is suggested in the introduction that this is the primary distinguishing aspect of spatial clustering from ordinary clustering… The authors do have a small paragraph on a spatial penalty (which is mostly unused throughout the paper) — but unless I misunderstand, the main algorithm seems to treat spatial coordinates and features x as if a combined feature vector.  Note: I think that it would *not* be so if in the Algorithm they used GPR([Xj], Yj) instead of GPR([Sj, Xj], Yj) as is done.

The introductory paragraph in Sec 2.2 seems like it must have a mistake somewhere.  s_i is meant to represent spatial coordinates, but later in the same paragraph, it says that its cluster index is j iff s_i is in S_j (where S_j is one of the regions that partition Omega?).  Do the authors mean to say that the collection of S_j partition R^2 instead?  Moreover, f_j is defined in this paragraph to be some unknown function on Omega, but everywhere else in the paper seems to suggest that it is actually treated as a function on (R^2 x Omega).

So going back to Algorithm 1, I would ask the authors to consider if the partition into clusters were defined instead over (R^2 x Omega), would that actually change anything about how Algorithm 1 should be defined?

Another related question that I have is — is there a way to frame this problem as an ordinary mixture of gaussian processes?  If so, can Algorithm 1 be viewed as a form of “hard-EM” for mixture estimation?  And if that is the case, would it make sense to try a soft-EM form of the same procedure?

Beyond whether this paper is truly a spatial clustering paper or not, the other major weakness is that the results are primarily conducted on synthetic data.  It would really help to have more than one “realistic” data set.  My concern is that GPs model such a general class of functions that I’m not sure its easy to separate clusters sometimes under such a general assumption.  In the North carolina example, the authors state that they use 3 clusters as that yielded the most interpretable results, but I would ask whether there was a more objective way to arrive at this conclusion.

---

### Note · Authors · 2024-06-30

I have read and agree with the venue's withdrawal policy on behalf of myself and my co-authors.